# STT3-dependent PD-L1 accumulation on cancer stem cells promotes immune evasion

Jung-Mao Hsu[1], Weiya Xia[1], Yi-Hsin Hsu[1], Li-Chuan Chan[1,2], Wen-Hsuan Yu[1,2], Jong-Ho Cha[1,3], Chun-Te Chen[1], Hsin-Wei Liao[1,2,4], Chu-Wei Kuo[5], Kay-Hooi Khoo[5], Jennifer L. Hsu[1,6], Chia-Wei Li[1], Seung-Oe Lim[1], Shih-Shin Chang[1,2], Yi-Chun Chen[1], Guo-xin Ren[7] & Mien-Chie Hung[1,2,6,8]

Enriched PD-L1 expression in cancer stem-like cells (CSCs) contributes to CSC immune evasion. However, the mechanisms underlying PD-L1 enrichment in CSCs remain unclear. Here, we demonstrate that epithelial–mesenchymal transition (EMT) enriches PD-L1 in CSCs by the EMT/β-catenin/STT3/PD-L1 signaling axis, in which EMT transcriptionally induces N-glycosyltransferase STT3 through β-catenin, and subsequent STT3-dependent PD-L1 N-glycosylation stabilizes and upregulates PD-L1. The axis is also utilized by the general cancer cell population, but it has much more profound effect on CSCs as EMT induces more STT3 in CSCs than in non-CSCs. We further identify a non-canonical mesenchymal–epithelial transition (MET) activity of etoposide, which suppresses the EMT/β-catenin/STT3/PD-L1 axis through TOP2B degradation-dependent nuclear β-catenin reduction, leading to PD-L1 downregulation of CSCs and non-CSCs and sensitization of cancer cells to anti-Tim-3 therapy. Together, our results link MET to PD-L1 stabilization through glycosylation regulation and reveal it as a potential strategy to enhance cancer immunotherapy efficacy.

[1] Department of Molecular and Cellular Oncology, The University of Texas MD Anderson Cancer Center, Houston, TX 77030, USA. [2] The University of Texas Graduate School of Biomedical Sciences at Houston, Houston, TX 77030, USA. [3] Tumor Microenvironment Global Core Research Center, College of Pharmacy, Seoul National University, Seoul 151-742, Korea. [4] Center for Systems Biology, Massachusetts General Hospital Research Institute, Harvard Medical School, Boston, MA 02114, USA. [5] Institute of Biological Chemistry, Academia Sinica, Taipei 115, Taiwan. [6] Graduate Institute of Biomedical Sciences and Center for Molecular Medicine, China Medical University, Taichung 404, Taiwan. [7] Department of Oral and Maxillofacial, Head and Neck Oncology, Affiliated 9th People's Hospital, School of Medicine, Shanghai Jiaotong University, Shanghai 200011, China. [8] Department of Biotechnology, Asia University, Taichung 413, Taiwan. These authors contributed equally: Weiya Xia, Yi-Hsin Hsu, Li-Chuan Chan. Correspondence and requests for materials should be addressed to M.-C.H. (email: mhung@mdanderson.org)

Cancer cells express various molecules that deliver either stimulatory or inhibitory signals during direct physical contacts with tumor-infiltrating lymphocytes (TILs). The balance of these opposing signals regulates the amplitude and quality of TIL response, and aberrant activation of the inhibitory signals, also known as immune checkpoints, is a mechanism utilized by cancer cells to evade immune attacks[1]. The programmed cell death protein-1 (PD-1)/programmed death-ligand 1 (PD-L1) axis is one of the major immune checkpoints identified to date in which binding of ligand PD-L1 (on antigen-presenting cells or cancer cells) to receptor PD-1 (on TILs) transmits inhibitory signals to suppress the activation, expansion, and acquisition of effector functions of TILs, especially CD8[+] cytotoxic T cells[1,2]. Evasion of immune surveillance through upregulation of PD-L1 expression is observed in many cancer types[1,3], and therapeutic antibodies against PD-1 or PD-L1 have shown promising outcomes[1,4–6]. However, only a proportion of patients respond to the treatments. Thus, furthering our understanding of the regulation underlying PD-L1 expression may identify biomarkers or lead to new combinational strategies to improve the efficacy of PD-1/PD-L1 blockade therapies[7,8].

Multiple signaling pathways via transcriptional control have been reported to regulate cancer cell PD-L1 expression[9,10]. Recently, our group demonstrated that PD-L1 is also subjected to protein N-glycosylation, which is critical in maintaining PD-L1 protein stability through antagonizing β-TrCP-dependent proteasome degradation of PD-L1[11]. However, the key components responsible for PD-L1 N-glycosylation remain to be explored.

Cancer stem-like cells (CSCs), also known as tumor-initiating cells, are a minor subpopulation of tumor cells and play important roles in tumor initiation, progression, and drug resistance[12,13]. CSCs are more resistant to immunological control compared with non-CSCs, and cancer immunosurveillance enriches a subpopulation of cancer cells with stem-like properties[14]. CSC immune evasion is critical for CSCs to sustain the tumorigenic process[15,16]. Previous studies have shown that CSCs express low levels of molecules involved in processing and presenting tumor antigens to T cell receptors (TCRs), a crucial stimulatory signal to T-cell response[15,16]. Consequently, CSCs escape from recognition by anti-tumor immunity. Interestingly, accumulating evidence indicates that CSCs also actively suppress T-cell activation[17,18]. Recent studies further suggested that enriched PD-L1 in CSCs may contribute to CSC immune evasion[19]. Although many signaling pathways have been linked to PD-L1 regulation in the general cancer cell population, which is composed largely of non-CSCs[9,10], the regulatory mechanisms contributing to the enriched PD-L1 expression in the CSC populations remain unclear.

In the current study, we investigate the underlying mechanism conferring enriched PD-L1 expression in CSCs and report a mechanism-driven approach to overcome CSC immune evasion.

## Results

**Epithelial–mesenchymal transition (EMT) enriches PD-L1 protein expression in CSCs.** Enriched PD-L1 expression in CSCs has been suggested to facilitate CSC immune evasion in lung[20] and head and neck[19] cancers. We first validated whether enriched PD-L1 expression is observed in the CSC populations of breast cancer cells and contributes to breast CSC immune evasion. Compared with non-CSC populations, enriched PD-L1 expression was observed in CSC populations (CD44[+]CD24[−/low] population in human breast cancer[21] and CD44[+]CD24[+]ALDH1[+] population in mouse breast cancer[22]) of multiple triple-negative breast cancer (TNBC) cell lines (Supplementary Fig. 1a–c). We then compared the sensitivity of CSC and non-CSC populations to

peripheral blood mononuclear cell (PBMC)-mediated cancer cell killing in vitro in the presence or absence of PD-L1. As expected, CSCs were more resistant to PBMC-mediated killing in vitro as shown by reduced level of cleaved caspase 3. However, following PD-L1 knockout, both CSC and non-CSC populations showed similar levels of cleaved caspase 3 (Supplementary Fig. 1d), suggesting that the enhanced PD-L1 expression in CSCs contributes to CSC resistance to PBMC-mediated killing in vitro in our breast cancer model system.

The above-mentioned results prompted us to ask how the enriched PD-L1 expression of CSCs is regulated. In the general cell population, EMT is known to regulate PD-L1[23]. CSCs comprise only a small portion of the entire cell population and frequently exhibit differential response to extracellular stimuli, e.g., therapeutic agents and growth factors, compared with non-CSC populations[24,25]. However, it is unclear whether the enriched PD-L1 expression of CSCs may also be regulated in response to EMT. Consistent with earlier findings in lung cancer cells[23], in the general cell population, PD-L1 was upregulated in breast epithelial cells MCF-10A undergoing EMT driven by TGF-β or RasV12 (Fig. 1a, c). In further comparisons of PD-L1 induction in the stem-like cell (SC) and non-SC populations using flow cytometric analysis, we noticed that while EMT upregulated PD-L1 of both populations, EMT led to a more robust PD-L1 induction in the SC populations (10–13-fold induction) than in the non-SC populations (3–4-fold induction) (Fig. 1b, d). To validate EMT-mediated PD-L1 induction in SCs, we further compared PD-L1 protein levels in spheres grown from parental cells and from cells undergoing EMT. Spheres cultured from non-adherent conditions are derived only from self-renewing cells and enriched in stem-like populations[26,27]. Consistently, elevated PD-L1 expression was observed in spheres grown from cells undergoing EMT (Fig. 1e, f). Together, these findings suggest that EMT induces more PD-L1 protein in CSCs than in non-CSCs.

PD-L1 is known to be transcriptionally upregulated upon EMT in the general cancer cell population[23]. When we compared PD-L1 (CD274) mRNA induction between SC and non-SC populations, however, there was no significant difference in the PD-L1 (CD274) mRNA fold change between the two populations (Fig. 1g, h and Supplementary Fig. 1a–c), suggesting that transcriptional upregulation was not the primary mechanism by which EMT enriches PD-L1 in CSCs over non-CSCs.

**STT3 is critical for EMT-mediated PD-L1 protein induction.** PD-L1 protein is modified by N-glycosylation[28,29]. Glycosylated PD-L1 is around 45 kDa (Supplementary Fig. 2a) and recognized by concanavalin A (ConA) lectin (Supplementary Fig. 2b), which recognizes N-glycans through its α-mannose-binding specificity[30]. Depletion of PD-L1 glycosylation by N-glycosidase F (PNGase F) treatment or by glycosylation-site mutagenesis (4NQ mutant)[11] caused a molecular weight shift of PD-L1 from 45 kDa to 33 kDa and abolished its ConA lectin binding (Supplementary Fig. 2a, b). N-glycosylation of PD-L1 is critical for PD-L1 protein stabilization by preventing PD-L1 from ubiquitin/proteasome-mediated degradation[11]. Mass spectrometry analysis showed that the N-glycan site occupancy (%) of PD-L1 N-glycosylation sites was upregulated upon EMT (Supplementary Fig. 3), implying a potential involvement of PD-L1 N-glycosylation in EMT-mediated PD-L1 induction.

To identify the N-glycosyltransferase(s) responsible for PD-L1 N-glycosylation and characterize the role of PD-L1 N-glycosylation in PD-L1 induction upon EMT, we scored the EMT status (a higher score indicates a more mesenchymal-like signature) of samples from The Cancer Genome Atlas (TCGA) breast cancer

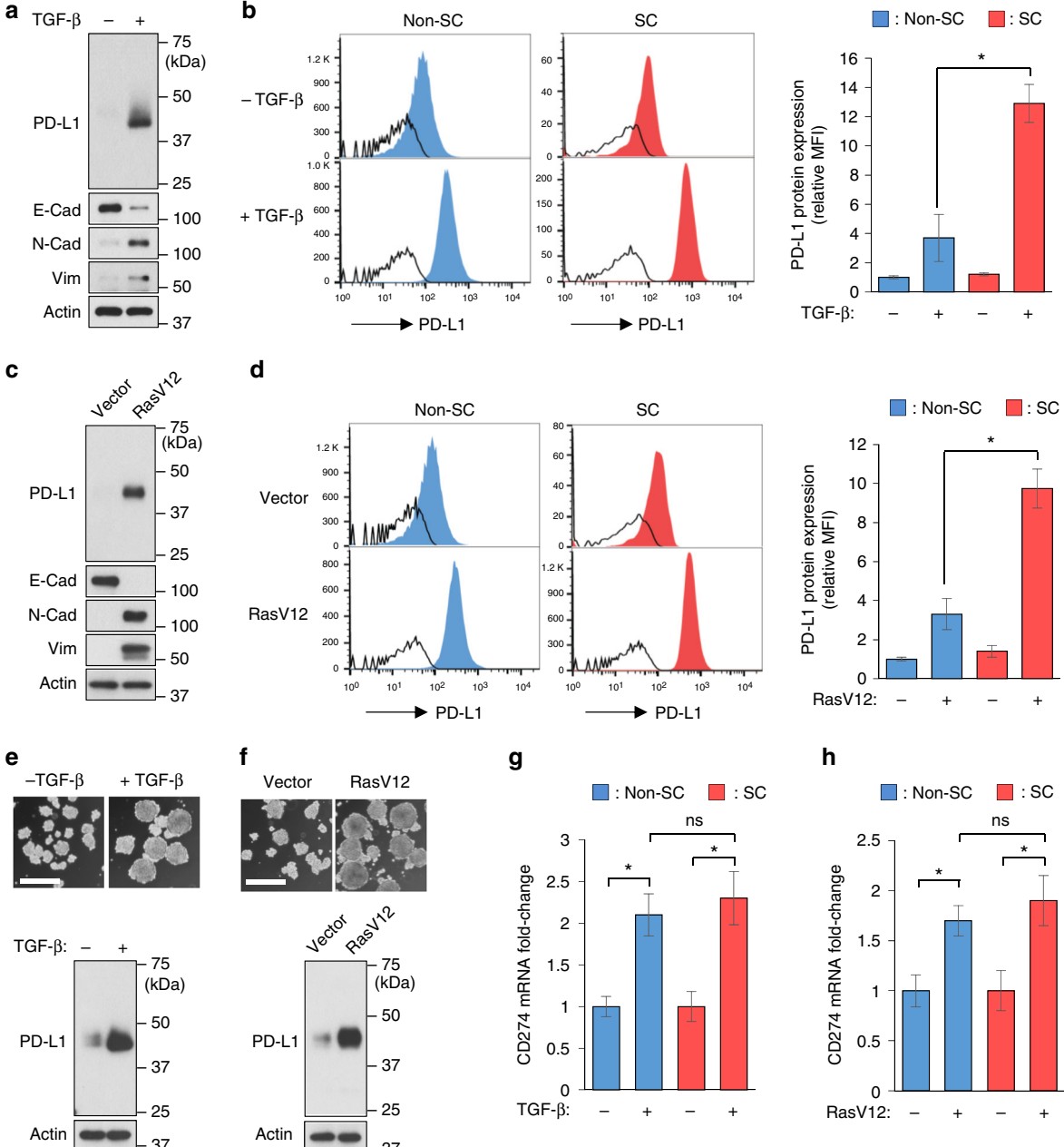

**Fig. 1** EMT induces a more robust upregulation of PD-L1 protein in CSCs than in non-CSCs without significant induction of PD-L1 mRNA levels. **a**, **c** Western blot analysis of PD-L1 and EMT markers in the general cell population of MCF-10A cells undergoing EMT driven by TGF-β (**a**) or RasV12 (**c**). E-Cad E-cadherin, N-Cad N-cadherin, Vim vimentin. **b**, **d** Flow cytometric analysis of PD-L1 protein expression in breast stem-like cell (SC) (CD44$^+$CD24$^{-/low}$, red histograms) and non-SC (non-CD44$^+$CD24$^{-/low}$, blue histograms) populations from control cells and EMT cells driven by TGF-β (**b**) or RasV12 (**d**). Open histograms represent isotype IgG negative control. The mean fluorescence intensity (MFI) of each cell population was quantified by FlowJo and compared. **e**, **f** Representative phase-contrast microscopy images and PD-L1 expression of spheres growing from control or EMT cells driven by TGF-β (**e**) or RasV12 (**f**). Scale bar, 200 μm. **g**, **h** qRT-PCR analysis comparing PD-L1 (CD274) mRNA induction levels between SC (CD44$^+$CD24$^{-/low}$) and non-SC (non-CD44$^+$CD24$^{-/low}$) populations upon TGF-β (**g**) or RasV12 (**h**) treatment. Error bars represent s.d. ($n = 3$). *$P < 0.05$; ns: non-significant, Student's $t$-test. See also Supplementary Fig. 1

dataset ($n = 1100$) (Supplementary Data 1)[31] and analyzed the correlations between N-glycosyltransferases and PD-L1 (CD274) mRNA expression, and EMT scores. In agreement with earlier findings[23], PD-L1 mRNA expression was positively correlated with EMT score (Supplementary Fig. 4a). Notably, both endoplasmic-reticulum (ER)-associated N-glycosyltransferase STT3 isoforms (including A and B two isoforms in mammalian cells) were also positively correlated with EMT score (Supplementary Fig. 4a). Moreover, in breast cancer cell lines and breast

cancer tissues ($n = 129$), the mRNA and protein expression levels of STT3 isoforms and PD-L1 were negatively correlated with E-cadherin expression (Supplementary Fig. 4b, c), a hallmark of epithelial traits. These findings implied a concomitant induction of the STT3 isoforms and PD-L1 in cells undergoing EMT. Indeed, along with PD-L1 induction, EMT driven by TGF-β or RasV12 upregulated STT3 isoforms at both mRNA and protein levels in the general cell population (Fig. 2a, b). To understand the roles of STT3 isoforms in PD-L1 induction, we first examined

the effects of exogenous STT3 isoforms on PD-L1 protein and mRNA levels in breast epithelial cells without TGF-β stimulation. The results showed that ectopic expression of either STT3 isoform alone was sufficient to induce PD-L1 protein levels, not mRNA levels (Fig. 2c). Next, we knocked down endogenous STT3 isoforms in breast epithelial cells and found that the loss of either isoform alone did not significantly impair PD-L1 induction under

TGF-β stimulation (Fig. 2d) while STT3A knockdown partially reduced the glycosylation status of PD-L1 (as indicated by lower PD-L1 molecular weight in lane 4 relative to lane 2). However, knocking down both STT3 isoforms significantly suppressed EMT-mediated PD-L1 induction in protein levels (lane 8, Fig. 2d), not mRNA levels (Fig. 2g) and only the ~33-kDa PD-L1 was induced (lane 8, longer exposure, Fig. 2d). Similar effects were

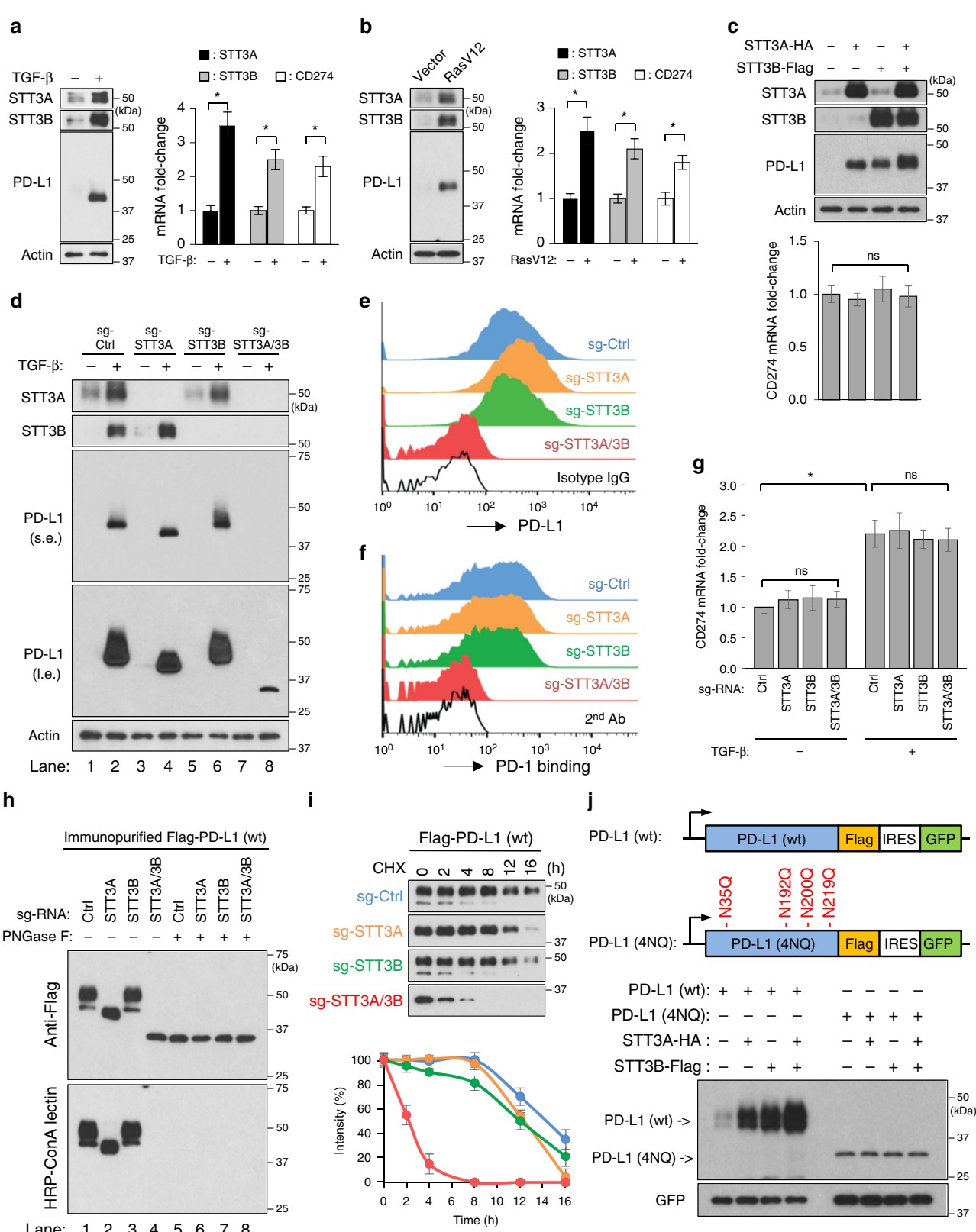

**Fig. 2** STT3-dependent PD-L1 glycosylation is sufficient and required for effective EMT-mediated PD-L1 induction in the general cell population. **a, b** Protein and mRNA induction of STT3A, STT3B, and PD-L1 (CD274) in the general cell population of MCF-10A cells undergoing EMT driven by TGF-β (**a**) or RasV12 (**b**). **c** Effect of ectopic expression of STT3 isoforms on endogenous PD-L1 protein and mRNA induction in the general cell population of MCF-10A cells without TGF-β treatment. **d** Influence of endogenous STT3 isoforms knockdown on TGF-β-mediated PD-L1 protein induction and PD-L1 molecular weight; s. e. short exposure, l.e. long exposure. **e, f** Flow cytometric analysis comparing cell surface PD-L1 expression (**e**) and PD-1 binding abilities (**f**) between various sgRNA-transfected MCF-10A cells after TGF-β treatment. **g** qRT-PCR analysis of PD-L1 mRNA levels in STT3 knockdown MCF-10A cells upon TGF-β treatment. **h** ConA lectin binding assay analyzing the glycosylation status of PD-L1 protein purified from STT3 knockdown cells. **i** Cycloheximide (CHX) chase assay of PD-L1 protein turnover rates in STT3 knockdown cells. **j** Top: Schematic illustrating co-expression constructs of PD-L1 (wt or 4NQ) and green fluorescent protein (GFP) used to assay the protein expression status of PD-L1. GFP was used as an internal control for transfection efficiency and gene expression. IRES internal ribosome entry site. Bottom: effect of ectopic STT3 isoforms on the protein expression amounts of wild-type (wt) PD-L1 and glycosylation-site mutant (4NQ). Error bars represent s.d. ($n = 3$). *$P < 0.05$; ns: non-significant, Student's $t$-test. See also Supplementary Figs. 2–5

also observed in endogenous PD-L1 in mesenchymal-like TNBC cells (Supplementary Fig. 5a, d). Accompanied by impaired PD-L1 induction, STT3A/3B knockdown diminished the PD-1 binding ability of cancer cells (Fig. 2e, f and Supplementary Fig. 5b, e). Together, these data suggested that N-glycosyltransferases STT3 isoforms are sufficient and required for effective PD-L1 protein induction in the general cell population of mesenchymal-like cancer cells.

STT3A/3B had no appreciable effect on PD-L1 mRNA level (Fig. 2c, g and Supplementary Fig. 5c, f). Since the molecular weight of PD-L1 protein in STT3A/3B knockdown cells (Fig. 2d and Supplementary Fig. 5a, d) was closed to that of unglycosylated PD-L1 (Supplementary Fig. 2a, b) and PD-L1 glycosylation is critical for PD-L1 protein stability[11], we then interrogated whether STT3 may regulate PD-L1 induction through PD-L1 protein N-glycosylation and stability. The results showed that, in STT3A/3B knockdown cells, PD-L1 lost its binding ability with ConA lectin and exhibited a molecular weight that was similar to that of PNGase F-treated/unglycosylated PD-L1 (lane 4, Fig. 2h), indicating that PD-L1 is unglycosylated in STT3A/3B knockdown cells. As reported earlier[11], the half-life of unglycosylated PD-L1 in STT3A/3B knockdown cells was significantly shorted than that of glycosylated PD-L1 in the parental or STT3 single knockdown cells (Fig. 2i), suggesting that STT3 isoforms are critical for PD-L1 glycosylation and stabilization. In line with these findings, ectopic STT3 isoforms (Fig. 2j) or STT3A/3B knockdown (Supplementary Fig. 5g) affected protein expression of wild-type (wt) PD-L1, but not unglycosylated PD-L1 mutant (4NQ). Together, these results support a notion that the two STT3 isoforms regulate EMT-mediated PD-L1 induction through PD-L1 protein N-glycosylation and stabilization.

**EMT induced higher levels of STT3 in CSCs than in non-CSCs.** The above-mentioned results prompted us to further ask whether STT3 isoforms may contribute to EMT-mediated enriched PD-L1 expression in CSCs than in non-CSCs. To this end, we compared EMT-mediated STT3 induction between CSC and non-CSC populations. The results showed that, in breast epithelial cell MCF-10A, while EMT driven by TGF-β or RasV12 upregulated STT3 isoforms in both populations, significantly higher levels of STT3 were observed in SCs than in non-SCs at both mRNA and protein levels (Fig. 3a, b). Higher STT3 expression was also detected in CSCs than in non-CSCs of breast cancer cells with intrinsic mesenchymal-like phenotype (Fig. 3c). Moreover, knockdown of both STT3 isoforms suppressed PD-L1 induction in both CSC and non-CSC populations and diminished EMT-mediated enriched PD-L1 expression in CSCs (Fig. 3d), leading to sensitization of CSCs to PBMC-mediated cancer cell killing in vitro (Fig. 3e). These results suggested that EMT induces higher levels of STT3 in CSCs than in non-CSCs, leading to enriched PD-L1 expression of CSCs.

**EMT transcriptionally upregulates STT3 through β-catenin.** Since STT3-dependent PD-L1 N-glycosylation is critical for EMT-mediated PD-L1 induction in both CSC and non-CSC populations, we next asked how EMT induces STT3 isoforms. By searching the Encyclopedia of DNA Elements (ENCODE) database to determine whether any EMT-related transcription factors bind to the transcriptional regulatory regions (as defined by enriched histone H3K4Me3 and DNase Clusters[32,33]) of STT3 isoforms, we noticed a co-enrichment of H3K4Me3, DNase Clusters, and TCF4 (TCF7L2), a transcriptional transactivation partner of EMT transcription factor β-catenin[34,35], near the transcription start sites of STT3 isoforms (Fig. 4a, red arrow). Moreover, β-catenin (CTNNB1) was positively correlated with STT3 isoforms in TCGA breast cancer dataset ($n = 1100$; Fig. 4b) and nuclear active β-catenin levels were positively correlated with the protein expression levels of STT3 isoforms in breast cancer tissues ($n = 129$; Supplementary Fig. 4c). Furthermore, it is known that CSCs, compared with non-CSCs, exhibit higher levels of β-catenin signaling[36–38]. These findings together implied a potential regulatory role of β-catenin in STT3 expression. To this end, we showed that β-catenin activated the promoters of STT3 isoforms, and a dominant-negative mutant of TCF4 (TCF4-DN) or abolishment of TCF4 binding sites on the promoters diminished the activating effect of β-catenin (Fig. 4c) by luciferase reporter assays. Similar regulatory effects by β-catenin and TCF4-DN on endogenous STT3 isoforms were observed at both protein and mRNA levels (Fig. 4d), suggesting that β-catenin induces STT3 isoforms. Along with STT3 induction, PD-L1 was also induced by β-catenin and then suppressed by STT3A/3B knockdown (Fig. 4e), further confirming the crucial role of STT3 isoforms in PD-L1 induction. Additionally, β-catenin knockdown or TCF-DN suppressed STT3 isoforms and PD-L1 in TGF-β-treated epithelial cells (Fig. 4f) or in cells with intrinsic mesenchymal-like phenotype (Fig. 4g). Together, these data suggested that EMT induces STT3 isoforms, resulting in PD-L1 upregulation through β-catenin/TCF4-mediated transcriptional upregulation of both STT3 isoform genes.

**TOP2 poisons exhibit non-canonical anti-EMT activity.** The above results indicated that the EMT/β-catenin/STT3/PD-L1 signaling axis is critical for EMT-mediated PD-L1 induction in both CSC and non-CSC populations of mesenchymal-like cancer cells with a much more profound effect on the CSC population. They also raised an interesting possibility that anti-EMT may be a potential strategy to modulate PD-L1 expression and improve cancer immune eradication. However, EMT inhibitor is not available in the clinic. To this end, we searched for potential EMT inhibitors using a series of clinically used chemotherapeutic agents to reverse the EMT status of mesenchymal-like TNBC 4T1 cells. E-cadherin expression was assessed as a marker for EMT reversal. Interestingly, E-cadherin expression was upregulated by several TOP2 poisons, including daunoribucin,

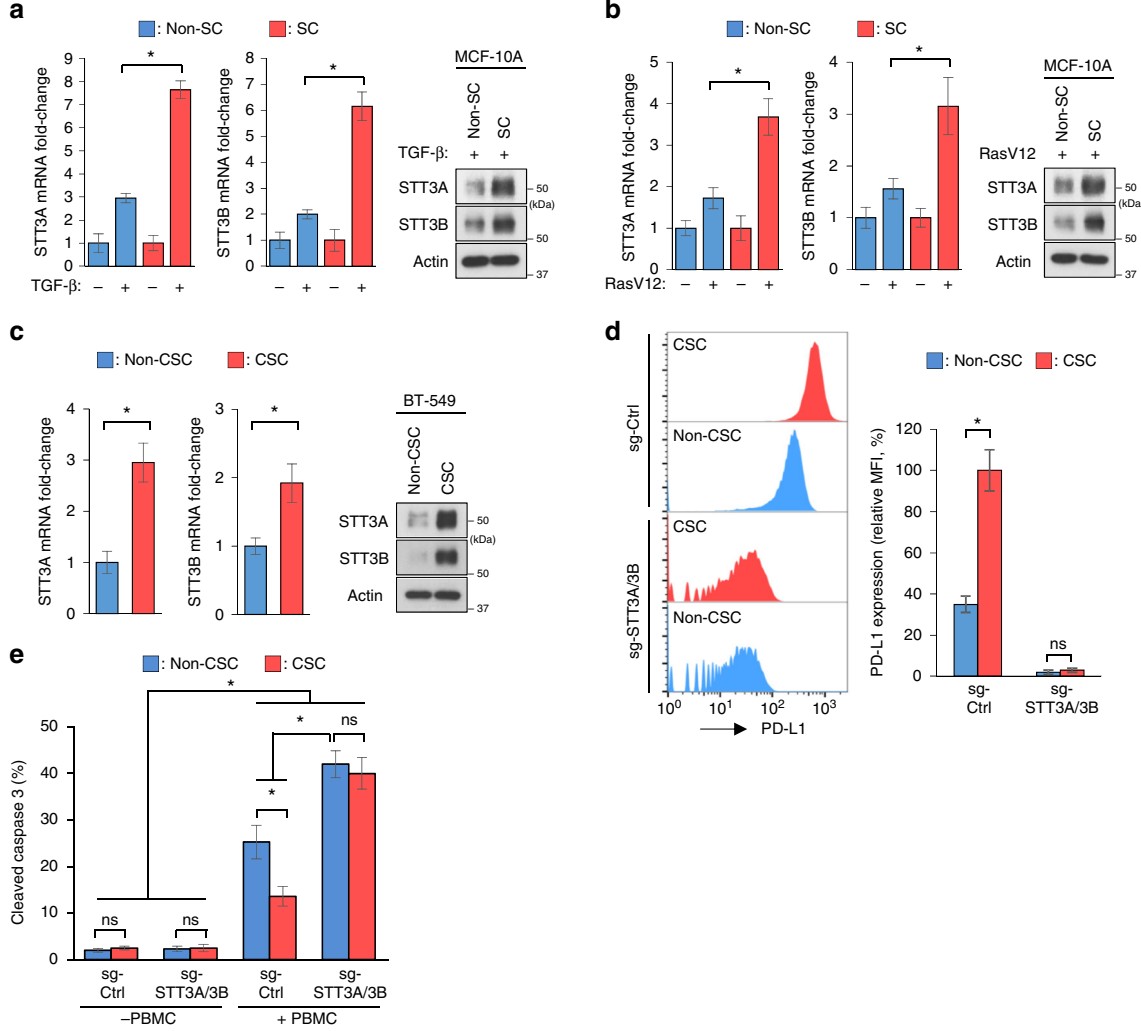

**Fig. 3** EMT induces higher levels of STT3 in CSCs than in non-CSCs, resulting in enriched PD-L1 in CSCs. **a**, **b** qRT-PCR and western blot analyses of STT3 induction in SCs (CD44[+]CD24[−/low] populations) and non-SCs (non-CD44[+]CD24[−/low] populations) of MCF-10A after TGF-β (**a**) or RasV12 (**b**) treatment. **c** qRT-PCR and western blot analyses of intrinsic STT3 expression in CSCs (CD44[+]CD24[−/low] populations) and non-CSCs (non-CD44[+]CD24[−/low] populations) of BT-549 cells. **d** Flow cytometric analysis of the effect of STT3A/3B knockdown on PD-L1 expression in CSCs and non-CSCs. **e** Efficacy of STT3A/3B knockdown in sensitizing CSCs to PBMC-mediated cancer cell killing. Error bars represent s.d. ($n = 3$). *$P < 0.05$; ns: non-significant, Student's $t$-test

doxorubicin, epirubicin, etoposide, and mitoxantrone (Fig. 5a, red highlighted). TOP2 poisons significantly reduced cell number similar to paclitaxel (Fig. 5b, e). While most of the resting cells were viable following treatment (Fig. 5c, f), only TOP2 poisons, not paclitaxel, induced mesenchymal–epithelial transition (MET), as indicated by the increased expression of E-cadherin in conjunction with reduced expression of N-cadherin and vimentin (Fig. 5d, g). MET induction by TOP2 poisons was achieved by short-course high-dose (Fig. 5b–d) or continuous low-dose (metronomic) treatment schedules (Fig. 5e–g). In addition to the changes in the expression of EMT molecular markers, TOP2 poison-treated cells exhibited epithelial-like morphology (Fig. 5h) and attenuated cell migration and invasion abilities (Fig. 5i, j). Together, these findings suggested that TOP2 poisons exhibit anti-EMT ability beyond their well-known cytotoxic activities.

**Reversal of EMT by etoposide downregulates PD-L1**. Among the TOP2 poisons tested (Fig. 5a, red highlighted), we chose etoposide as a candidate to inhibit the EMT/β-catenin/STT3/PD-L1 axis in both CSCs and non-CSCs for the following reasons: (1)

etoposide is one of the few cytotoxic compounds with CSC-targeting ability[39]; (2) etoposide treatment does not induce significant changes in the number of CD8[+] lymphocytes in vivo[40,41], the major immune cell population in the body's defense against cancers[1,7]; (3) etoposide, especially at lower doses, induces tumor-specific immunity in preclinical models[42,43]. We found that, along with MET induction, etoposide reduced STT3 isoforms and PD-L1 in the general cell population of mesenchymal-like TNBC cells (Fig. 6a and Supplementary Fig. 6a). Re-expression of exogenous STT3 isoforms diminished etoposide-induced PD-L1 downregulation (Fig. 6b), suggesting that etoposide suppresses PD-L1 through STT3 isoforms. In addition, etoposide decreased CSC (CD44[+]CD24[+]ALDH1[+]) frequency in the CSC populations (Fig. 6c, left), which is indicative of its CSC-targeting cytotoxic activity[39], and reduced CSCs' PD-L1 expression (Fig. 6c, right), suggesting that etoposide effectively modulates PD-L1 expression in both CSC and non-CSC populations of mesenchymal-like cancer cells.

CSCs have been show to play an essential role in cancer initiation and progression[12,13]. To further validate the PD-L1 modulation effect of etoposide on CSCs and exclude its

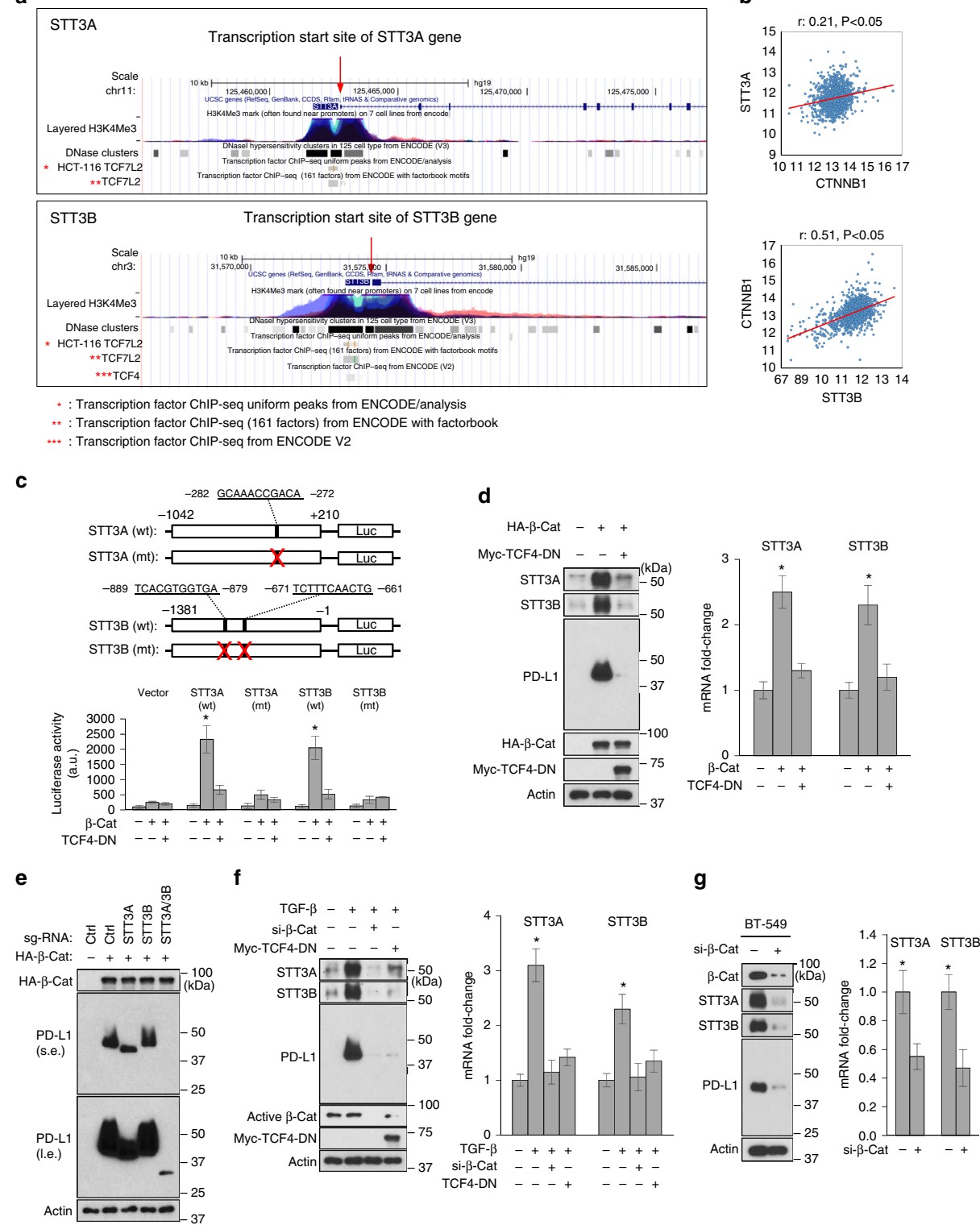

cytotoxicity, spheres were prepared from etoposide-treated cells in a treatment-free condition for comparison (Fig. 6d). Notably, the spheres derived from etoposide-treated cells expressed higher E-cadherin and lower N-cadherin and vimentin (Fig. 6e and Supplementary Fig. 6b), suggesting that etoposide confers CSCs a more epithelial-like status. Along with MET induction, etoposide attenuated glycosylated PD-L1 expression and PD-1 binding ability of sphere cells, and sensitized sphere cells to PBMC-

mediated cancer cell killing in vitro (Fig. 6e and Supplementary Fig. 6b). To specifically assay the functional effects of etoposide-induced MET and PD-L1 downregulation on CSC-mediated tumor initiation, growth, and immune modulation in the tumor microenvironment, serial dilutions of viable sphere cells in vitro cultured from etoposide-treated 4T1 cells were implanted into the mammary fat pads of syngeneic female BALB/c mice to form tumors. Results showed that spheres derived from etoposide-

**Fig. 4** EMT transcriptionally induces STT3 through β-catenin/TCF4. **a** ENCODE data display ChIP-seq signals for the occupancy of TCF4 (TCF7L2), histone H3K4Me3, and DNase Clusters around the transcription start sites of STT3A and STT3B. H3K4Me3 and DNase Clusters were used to define the transcriptional regulatory regions of STT3A and STT3B. These ENCODE data were generated by the ENCODE consortium and available on the Genome Browser at UCSC. **b** Pearson correlation analysis of β-catenin (CTNNB1) with STT3A and STT3B in TCGA breast cancer dataset ($n = 1100$). **c** Top: schematic presentation of the wild-type (wt) and TCF4-binding-site-mutated (mt) STT3 promoter-luciferase reporter constructs of STT3A ($-1042/+210$, related to the transcription start site) and STT3B ($-1381/-1$, related to the transcription start site). The TCF4 binding sites on the promoter regions of STT3A ($^{-282}$GCAAACCGACA$^{-272}$) or STT3B ($^{-889}$TCACGTGGTGA$^{-879}$, $^{-671}$TCTTTCAACTG$^{-661}$) are shown. Bottom: promoter luciferase activity in response to β-catenin (β-Cat) and TCF4 dominant-negative (TCF4-DN) mutant. **d** Effect of β-catenin (β-Cat) and TCF4-DN on the protein and mRNA expression of STT3 isoforms in MCF-10A cells. **e** Influence of STT3 knockdown on β-catenin (β-Cat)-mediated PD-L1 induction in MCF-10A cells; s.e. short exposure, l.e. long exposure. **f** Effect of β-catenin knockdown (si-β-Cat) and TCF4-DN on TGF-β-mediated PD-L1 induction in MCF-10A cells. **g** Effect of β-catenin knockdown (si-β-Cat) on the expression of STTA/B and PD-L1 in mesenchymal-like BT-549 cells. Error bars represent s.d. ($n = 3$). *$P < 0.05$, Student's $t$-test

treated cells exhibited impaired tumor-initiating capabilities with lower tumor incidence (Fig. 6f), delayed tumor onset (Fig. 6g), and attenuated tumor growth rate (Fig. 6h). Of note, immunostainings indicated that the spheres from etoposide-treated cells formed tumors with increased number of tumor-infiltrating cytotoxic T cells, as indicated by enhanced staining of CD8 and granzyme b (Fig. 6i). Collectively, these data suggested that etoposide inhibits PD-L1 expression of both CSC and non-CSC populations and attenuates the tumor-initiating ability and cancer cell-mediated immunosuppressive activity of CSCs.

**Etoposide synergizes with Tim-3 blockade therapy.** Since inhibition of the EMT/β-catenin/STT3/PD-L1 axis by etoposide downregulated PD-L1 expression in both CSC and non-CSC populations of mesenchymal-like cancer cells, we sought to determine whether etoposide would enhance the therapeutic efficacy of immune checkpoint blockade therapy. Co-expression of PD-1 and T cell immunoglobulin mucin-3 (Tim-3) on T cells has been linked to T cell exhaustion in cancers of animal models[44,45] and clinical patients[46–48]. Tim-3$^+$PD-1$^+$CD8$^+$ T cells represent the predominant subset of TILs and combined targeting of both Tim-3 and PD-1 pathways, rather than single targeting of either pathway, has been shown to effectively reverse T cell exhaustion and restore anti-tumor immunity[44,45]. Therefore, we asked whether etoposide may sensitize tumors to anti-Tim-3 therapy. To this end, two mesenchymal-like mice cancer syngeneic models, 4T1 and CT26, were treated with etoposide and/or anti-Tim-3 antibody (Fig. 7a). Tumor growth was monitored, and tumors resected from mice were subjected to analysis of the EMT status, STT3 expression and CSC frequency of tumor cells, and the activation status of tumor-infiltrating cytotoxic T cells. Results showed that etoposide alone reduced tumor burden (Fig. 7b and Supplementary Fig. 7a) and CSC frequency (Fig. 7c and Supplementary Fig. 7b). At the molecular level, etoposide induced MET (Fig. 7d and Supplementary Fig. 7c) and STT3 downregulation (Fig. 7e and Supplementary Fig. 7d) in tumor cells. Moreover, etoposide suppressed PD-L1 expression in both CSCs and non-CSCs with much more profound effect on CSCs (Fig. 7f and Supplementary Fig. 7e). However, etoposide alone did not significantly increase the population of activated CD8$^+$ T cells (Fig. 7g and Supplementary Fig. 7f), suggesting that etoposide-induced PD-L1 downregulation is not sufficient to reverse T cell dysfunction in the animal models and that the inhibitory effect of etoposide monotherapy on tumor burden and CSC frequency is likely attributed to the CSC-targeting cytotoxic activity of etoposide[39]. Interestingly, while anti-Tim-3 monotherapy did not exhibit significant effects on tumor growth (Fig. 7b and Supplementary Fig. 7a) or T cell exhaustion (Fig. 7g and Supplementary Fig. 7f) as reported previously[44,45], the combination of anti-Tim-3 antibody and etoposide, which downregulated PD-L1 in both CSC and non-CSC populations, enhanced the inhibitory efficacy

of Tim-3 blockade therapy on both tumor burden (Fig. 7b and Supplementary Fig. 7a) and CSC frequency (Fig. 7c and Supplementary Fig. 7b) with a concomitant induction of tumor-infiltrating activated CD8$^+$ T cell population (Fig. 7g and Supplementary Fig. 7f). These results suggested that etoposide enhances the therapeutic efficacy of Tim-3 blockade therapy against both CSC and non-CSC populations, and that combination of etoposide with Tim-3 blockage therapy may be an effective anti-cancer strategy.

**Etoposide reverses EMT via nuclear β-catenin downregulation.** Next, we asked how etoposide inhibits the EMT/β-catenin/STT3/PD-L1 axis. Mechanistically, TOP2 poisons induce cytotoxicity through inhibition of DNA strand passage activity of TOP2 by stabilizing the DNA-TOP2 complexes[49]. In addition to TOP2 poisons, two TOP2 catalytic inhibitors, ICRF-187 and ICRF-193, which directly target the ATPase domain of TOP2[49], also induced MET and downregulation of STT3 isoforms and PD-L1 (Supplementary Fig. 8a), further supporting the involvement of TOP2 in etoposide-mediated inhibition of the EMT/β-catenin/STT3/PD-L1 axis. There are two TOP2 isoforms in mammalian cells, TOP2A and TOP2B. Interestingly, silencing TOP2B alone, but not TOP2A, was sufficient to induce MET and downregulation of STT3 and PD-L1 (Fig. 8a), and confer cells an epithelial-like morphology (Fig. 8b). Moreover, etoposide still induced MET of the TOP2A-silenced cells but had no further effects on TOP2B-silenced cells (Fig. 8c). In addition, in line with the effects of etoposide treatment (Fig. 6e), TOP2B silencing induced MET and downregulated STT3 and PD-L1 in spheres (Fig. 8d, left), resulting in reduced PD-1 binding (Fig. 8d, upper right) and sensitization of sphere cells to PBMC-mediated cancer cell killing in vitro (Fig. 8d, bottom right). Together, these results implied that etoposide induced MET and suppressed STT3 and PD-L1 likely through TOP2B downregulation. Indeed, etoposide, ICRF-187 and ICRF-193 are known to induce TOP2B degradation through the proteasome pathway[50,51]. Blocking TOP2B degradation by proteasome inhibitor MG132 suppressed etoposide-mediated MET induction and downregulation of STT3 and PD-L1 (Fig. 8e, lane 4 and Supplementary Fig. 8b, c, lane 4), indicating that TOP2B degradation is critical for etoposide to induce MET and downregulate STT3 and PD-L1.

β-catenin transcriptionally activates STT3 isoforms, leading to PD-L1 induction in the EMT/β-catenin/STT3/PD-L1 axis (Fig. 4). To explore the linkage between TOP2B degradation and β-catenin, we examined the level of β-catenin in the nucleus, where TOP2B and transcriptionally active β-catenin are located, under TOP2B downregulation. As shown, although TOP2B downregulation led to an overall accumulation of total and non-phosphorylated (active) form of β-catenin (Fig. 8f), nuclear β-catenin was reduced (Fig. 8g, lane 8), which is expected to suppress the gene transactivation activity of β-catenin. These

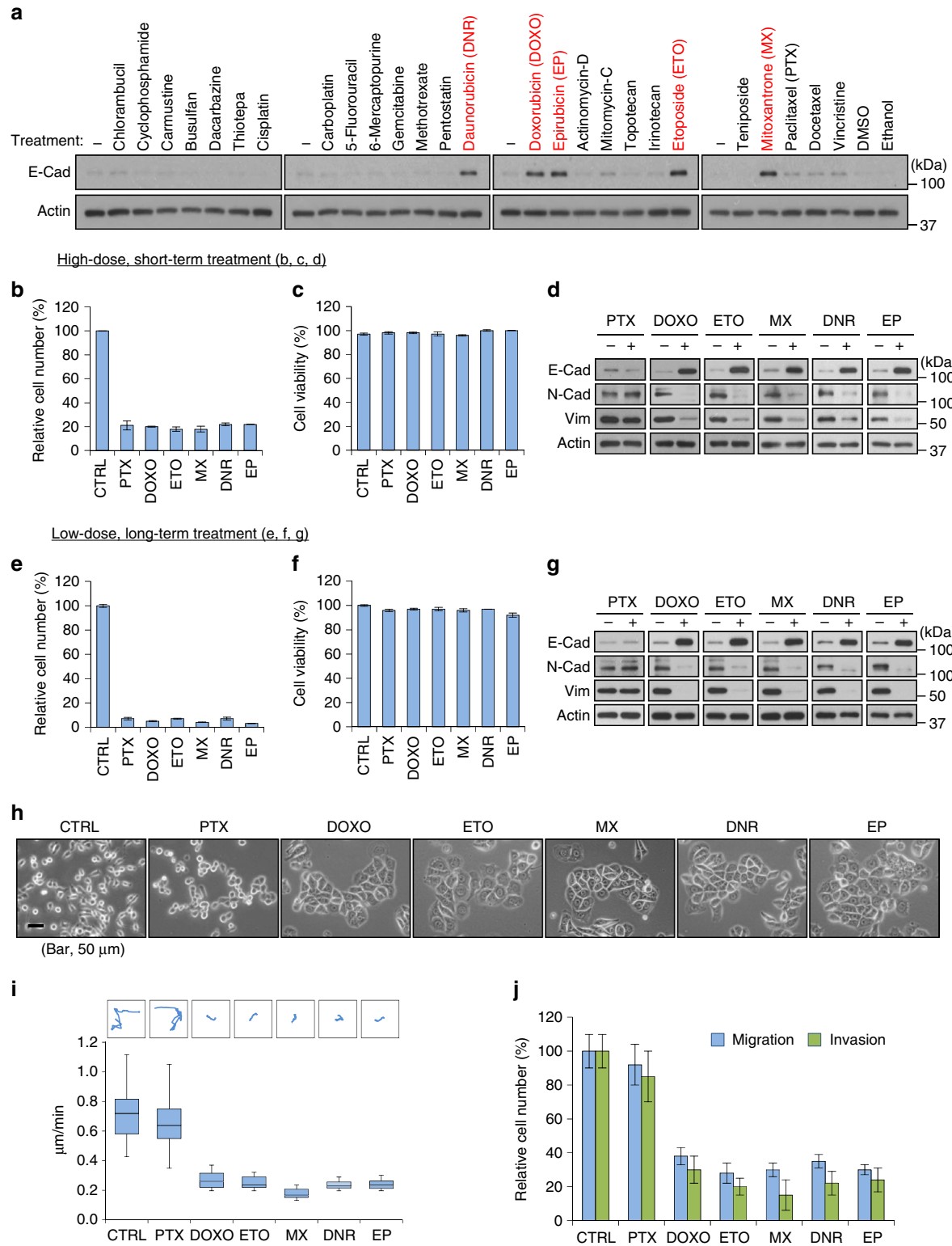

results implied that nuclear β-catenin downregulation may be the mechanism by which etoposide-mediated TOP2B degradation inhibits the EMT/β-catenin/STT3/PD-L1 axis. As expected, β-catenin knockdown, similar to etoposide, suppressed STT3 and PD-L1 in mesenchymal cancer cells (Fig. 8h, lanes 2 and 3) and rendered cells insensitive to etoposide or sg-TOP2B treatment (Fig. 8h, lanes 4 and 8), supporting the notion that etoposide-mediated TOP2B degradation and subsequent inhibition of the

EMT/β-catenin/STT3/PD-L1 axis is likely through downregulation of nuclear β-catenin. In parallel, we observed that TOP2B associated with β-catenin in the nucleus (Fig. 8i, j). Therefore, we rationalized that binding of β-catenin to TOP2B may facilitate nuclear localization of β-catenin and that etoposide degrades TOP2B and subsequently induces cytosolic translocation of nuclear β-catenin, leading to MET and downregulation of STT3 and PD-L1. Elevated E-cadherin stabilizes and traps β-catenin in

**Fig. 5** DNA topoisomerase II (TOP2) poisons induce mesenchymal–epithelial transition (MET). **a** Western blot analysis of epithelial marker E-cadherin (E-Cad) induction in mesenchymal-like 4T1 cells treated with indicated agents for 24 h (please check Supplementary Table 1 for compound concentrations). Compounds that significantly induce E-cadherin are labeled in red. **b** Relative cell number after treatment of indicated agents at a high-dose, short-term schedule. CTRL control, PTX paclitaxel (1 μM, 24 h), DOXO doxorubicin (2 μM, 24 h), ETO etoposide (10 μM, 24 h), MX mitoxantrone (2 μM, 24 h), DNR daunorubicin (1 μM, 24 h), EP epirubicin (1 μM, 24 h). **c** Cell viability of resting cells after treatment of indicated agents at a high-dose, short-term schedule. Cell viability was analyzed by trypan blue exclusion. **d** Western blot analysis of EMT markers (E-cadherin, N-cadherin, Vimentin) in resting cells after treatment of indicated agents at a high-dose, short-term schedule. **e** Relative cell number after treatment of indicated agents at a metronomic (low-dose, long-term) schedule. CTRL control, PTX paclitaxel (0.02 μM, 96 h), DOXO doxorubicin (0.05 μM, 96 h), ETO etoposide (0.2 μM, 96 h), MX mitoxantrone (0.05 μM, 96 h), DNR daunorubicin (0.02 μM, 96 h), EP epirubicin (0.02 μM, 96 h). **f** Cell viability of resting cells after treatment of indicated agents at a metronomic (low-dose, long-term) schedule. **g** Western blot analysis of EMT markers (E-cadherin, N-cadherin, Vimentin) in resting cells after treatment of indicated agents at a metronomic (low-dose, long-term) schedule. **h** Representative phase-contrast microscopy images of 4T1 cells treated with indicated agents. Scale bar, 50 μm. **i** Representative trajectories and quantification of motility speed of 4T1 cells treated with indicated agents. The box plots represent sample maximum (upper end of whisker), upper quartile (top of box), median (band in the box), lower quartile (bottom of box), and sample minimum (lower end of whisker). **j** Migration and invasion assays of 4T1 cells treated with indicated agents. All error bars represent s.d. ($n = 3$)

the cell membrane region, further preventing its nuclear translocation and gene transactivation[52]. Collectively, these data suggested that etoposide inhibits the EMT/β-catenin/STT3/PD-L1 axis through TOP2B degradation-dependent nuclear β-catenin downregulation.

## Discussion

Our results identified a mechanism by which CSCs evade immunosurveillance through enriched PD-L1 induction following EMT. Although EMT has been reported to epigenetically control PD-L1 through microRNA-200 in the general cell population[23], we found that this regulatory mechanism is not responsible for enriched PD-L1 expression of CSCs. We further demonstrated that EMT induces PD-L1 expression via the EMT/β-catenin/STT3/PD-L1 signaling axis (Supplementary Fig. 9) in which STT3 isoforms are sufficient and required for PD-L1 induction through regulating PD-L1 glycosylation and stabilization. This axis is not only critical for the effective PD-L1 induction in the general cell population of mesenchymal-like cancer cells but also required for enriched PD-L1 expression of CSCs as EMT induces higher levels of STT3 in CSCs. Therefore, the current study revealed how CSCs express higher PD-L1 than non-CSCs and suggested that the EMT/β-catenin/STT3/PD-L1 axis as a potential therapeutic target to downregulate PD-L1 of both CSC and non-CSC populations and overcome cancer immune evasion of mesenchymal-like cancer cells.

ER-associated N-glycosyltransferases, STT3A and STT3B, are required for PD-L1 N-glycosylation and stabilization. The STT3 isoforms are the catalytic subunits of oligosaccharyltransferase complex, which initiate N-glycosylation by catalyzing the transfer of a 14-sugar core glycan from dolichol to the asparagines of substrates[53,54]. Early studies have revealed a model by which the two STT3 isoforms act sequentially on polypeptides to maximize the efficiency of N-glycosylation[55]. The complementary roles of the STT3 isoforms in N-glycan addition may explain our results that upregulation of either STT3 isoform is sufficient to glycosylate and stabilize PD-L1 upon EMT. In addition to STT3-mediated incorporation of core glycan, our recent study also showed that β-1,3-Nacetylglucosaminyl transferase (B3GNT3)-mediated poly-LacNAc moiety on N192 and N200 glycosylation sites of PD-L1 is critical for PD-L1 binding with PD-1[56], suggesting that the distal portions of N-glycan on PD-L1 glycosylation sites are also functionally important in PD-L1 activity.

Etoposide is a commonly used cytotoxic anti-cancer agent, and previous studies demonstrated that it exhibits selective specificity against the mesenchymal/CSC population[39]. In our study, we found that etoposide treatment in mesenchymal-like cancer cells resulted in a cell population with epithelial phenotype. Although this observation could be interpreted as etoposide selectively

eliminating the mesenchymal/CSC subpopulation, this possibility was excluded from our model system by the results showing that other non-mesenchymal/CSC-specific TOP2B poisons/inhibitors also enriched epithelial-like cells. We further demonstrated that etoposide conferred cells epithelial-like phenotypes by reversing EMT through a TOP2B degradation-dependent mechanism, suggesting a non-canonical anti-EMT activity of etoposide in addition to its well-known cytotoxic activity. This finding may broaden the range of clinical applications of etoposide.

Etoposide was previously shown to induce tumor-specific immunity in which CD8[+] cytotoxic T cells played an essential role[42,43,57]. Notably, inoculation of mice with in vitro etoposide-treated cancer cells is sufficient to induce anti-tumor immunity, implying etoposide elicits tumor-specific immunity by inducing some modifications on cancer cells[42,43,57]; however, the underlying mechanisms are not fully understood. Here, we showed that reversal of EMT by etoposide led to PD-L1 downregulation in both CSC and non-CSC populations, suggesting a mechanism by which etoposide elicits anti-tumor immunity.

Although our results demonstrated that etoposide suppresses PD-L1, etoposide monotherapy does not achieve the same efficacy as well as durability in cancer therapy as anti-PD-1/PD-L1 antibodies in the clinic. This is likely attributed to the partial rather than complete downregulation of PD-L1 by etoposide, suggesting that etoposide monotherapy is not comparable to anti-PD-1/PD-L1 antibodies in PD-1/PD-L1 immune checkpoint modulation. However, etoposide may be applied in combined with immunotherapies. Our results from preclinical animal models demonstrated that etoposide synergizes with Tim-3 blockade therapy. Recently, etoposide also demonstrated therapeutic synergy with antibodies against cytotoxic T-lymphocyte-associated protein 4 (CTLA-4)[58]. Altogether, these findings suggest that etoposide, a conventional chemotherapeutic drug may provide additional benefits to cancer patients undergoing immunotherapies.

## Methods

**Cell culture**. MDA-MB-231, BT-549, Hs 578T, SK-BR-3, MDA-MB-436, MDA-MB-361, T-47D, ZR-75-1, BT-474, MCF7, MCF-10A, CT26, and 4T1 were purchased from American Type Culture Collection (ATCC). MCF-10A/RasV12 stable transfectant was established by our group previously[59]. MDA-MB-231, BT-549, Hs 578T, SK-BR-3, MDA-MB-436, MDA-MB-361, T-47D, ZR-75-1, BT-474, and MCF7 were cultured in DMEM/F-12 with 10% FBS. 4T1 and CT26 were cultured in RPMI-1640 with 10% FBS. MCF-10A and MCF-10A/RasV12 were cultured in DMEM/F12 medium supplemented with 5% horse serum, 10 mg ml$^{-1}$ insulin, 20 ng ml$^{-1}$ epidermal growth factor (EGF), 100 ng ml$^{-1}$ cholera toxin, and 500 ng ml$^{-1}$ hydrocortisone. To induce EMT of MCF-10A, the cells were treated with TGF-β (5 ng ml$^{-1}$, Invitrogen) for 10 days. All cell lines were characterized as mycoplasma negative and validated by STR DNA fingerprinting using the AmpFLSTR Identifiler kit (ThermoFisher) according to manufacturer's instructions. The STR profiles were compared with known ATCC fingerprints [https://www.atcc.org/] and with

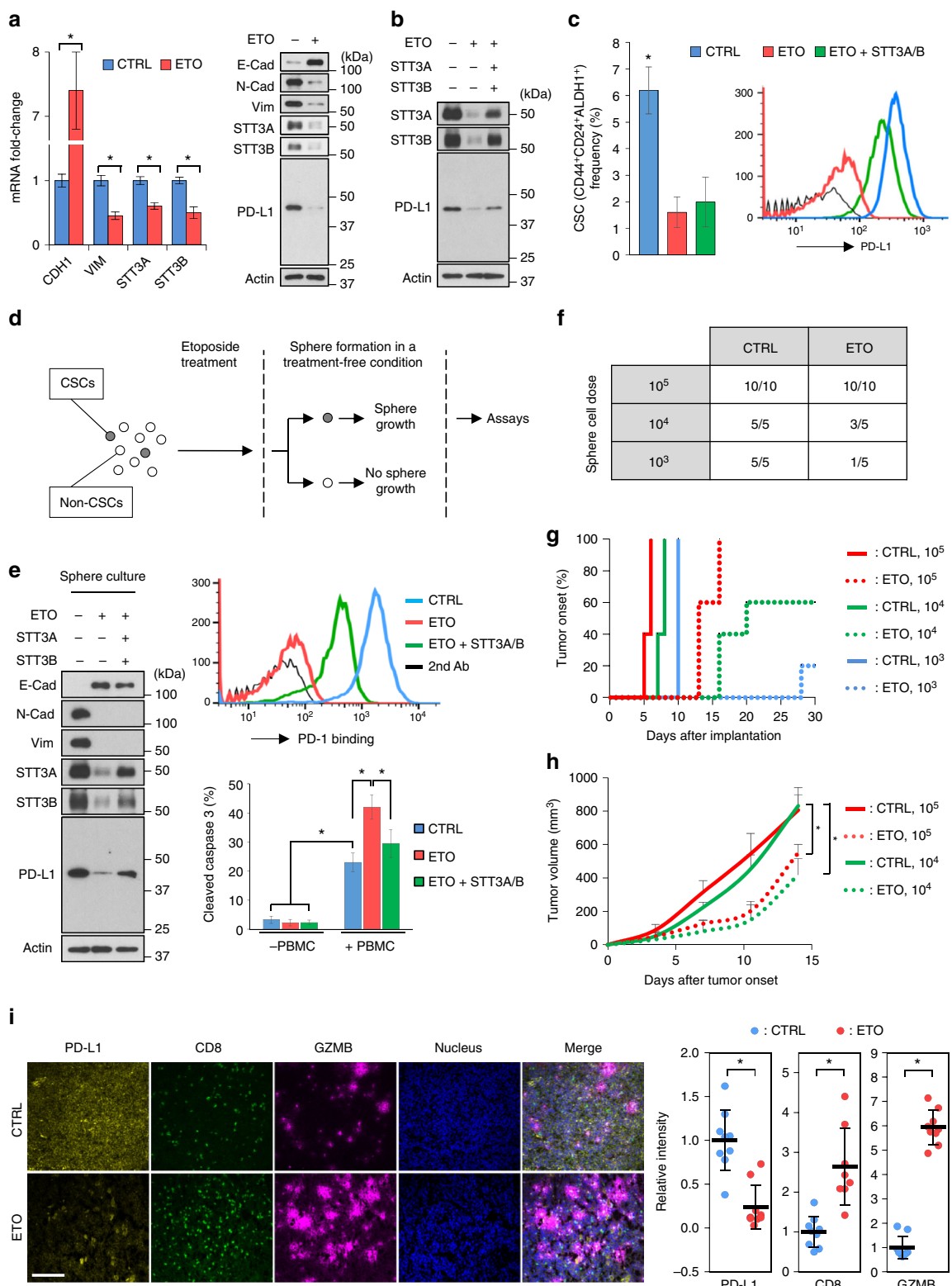

the Cell Line Integrated Molecular Authentication database (CLIMA) version 0.1.200808 [http://bioinformatics.istge.it/clima/][60]. The STR profiles matched the known DNA fingerprints or were unique.

**Reagents**. The following reagents were purchased from Sigma: MG-132, chlorambucil, cyclophosphamide, carmustine, busulfan, dacarbazine, thiotepa, cisplatin, 5-fluorouracil, 6-mercaptopurine, gemcitabine, methotrexate, doxorubicin, epirubicin, actinomycin-D, mitomycin-C, topotecan, irinotecan, etoposide, mitoxantrone, paclitaxel, docetaxel, and vincristine. The following reagents were purchased

from Calbiochem: carboplatin, pentostatin, and daunorubicin. The following reagents were purchased from Santa Cruz: ICRF-187, ICRF-193, and teniposide. The following reagents were purchased from MBL International Corporation: Z-VAD-FMK.

**Sphere assay**. Cells were dissociated with trypsin, washed, and cell viability was analyzed by trypan blue exclusion. Single-cell suspensions were plated (10,000 viable cells/well) in 6-well ultra-low attachment plates (Corning) in defined serum-free medium composed of DMEM/F-12, B27 supplement (Invitrogen), 20 ng ml$^{-1}$

**Fig. 6** Etoposide inhibits the EMT/β-catenin/STT3/PD-L1 axis and downregulates PD-L1 of CSC and non-CSC populations. **a** Effect of etoposide (ETO) on the protein and mRNA expression of EMT markers, STT3, and PD-L1 in the general cell population of 4T1 cells. E-Cad E-cadherin, N-Cad N-cadherin, Vim vimentin. **b** Effect of etoposide and exogenous STT3A/B on PD-L1 expression in the general cell population of 4T1 cells. **c** Flow cytometric analysis of the influence of etoposide and exogenous STT3A/B on the frequency (left) and PD-L1 expression (right) of 4T1 CSC (CD44$^+$CD24$^+$ALDH1$^+$) populations. Open histograms represent isotype IgG negative control. **d** Experimental workflow of preparing spheres derived from etoposide-treated 4T1 cells. **e** Western blot analysis (left), PD-1 binding assay (upper right), and in vitro PBMC-mediated tumor cell killing assay (bottom right) of tumorspheres derived from 4T1 cells treated with etoposide and exogenous STT3A/B. **f–h** Tumor-seeding ability (**f**), tumor onset curve (**g**), and tumor growth curve (**h**) of tumorspheres cultured from etoposide-treated 4T1 cells. **i** Representative images of tumors staining with PD-L1, CD8, granzyme b (GZMB), and Hoechst. Scale bar, 200 μm. The intensity of immunofluorescence signal was quantified from nine core biopsy sections, normalized relative to the control group, and shown as means ± s.d. Other error bars represent s.d. ($n = 3$). *$P < 0.05$, Student's $t$-test. See also Supplementary Fig. 6

recombinant bFGF (basic fibroblast growth factor; BD Biosciences), and 20 ng ml$^{-1}$ EGF. After 10-day culture, the number of spheres (size >50 μm for MDA-MB-231; size >100 μm for 4T1) were counted. Then, the spheres were harvested for western blotting or dissociated to single-cell suspensions for secondary sphere formation, animal experiments, or PD-1 binding assay. For compound-treatment experiments, the cells were treated with compounds under regular monolayer culture conditions and subjected to regular sphere culture in a compound-free condition.

**Protein extraction, western blotting, and immunoprecipitation**. Whole-cell extracts of monolayer-cultured or sphere cells were prepared by lysing the cells in a lysis buffer containing 50 mM Tris-HCl (pH 7.6), 150 mM NaCl, 1% NP-40, 1% sodium deoxycholate, 0.1% SDS, and 1× protease inhibitor (Roche) freshly added before lysis. Cell fractionations were prepared using ProteoExtract Subcellular Proteome Extraction Kit (Millipore). To perform immunoblotting, cell lysates were subjected to SDS-PAGE, transferred onto PVDF (Bio-Rad), and followed by western blot analysis using indicated antibodies. Quantifications of immunoblotting signal intensity were performed using ImageJ (NIH). To perform immunoprecipitation, cell lysates were incubated with antibodies with gentle rocking overnight at 4 °C and then incubated with protein A/G agarose beads (Sigma) for 3 h. After washing with lysis buffer, immunoprecipitation products were subjected to immunoblotting analysis. Antibodies used for immunoblotting and immunoprecipitation were as follows: E-cadherin (BD Biosciences, #610182, 1:10000), N-cadherin (BD Biosciences, #610921, 1:5000), vimentin (abcam, #ab45939, 1:5000), actin (Sigma, #A2066, 1:10000), PD-L1 (Cell Signaling, #13684, 1:1000; proteintech, #66248, 1:1000), STT3A (Santa Cruz, #SC-100796, 1:1000), STT3B (abcam, #ab122351, 1:500), TOP2A (Cell Signaling, #4733, 1:1000), TOP2B (R&D Systems, #MAB6348, 1:1000), Histone H3 (Cell Signaling, #3638, 1:2000), β-catenin (Cell Signaling, #8480, 1:2000), non-phospho (active) β-catenin (Cell Signaling, #8814, 1:2000). Uncropped scans of the most important western blots are shown in Supplementary Fig. 10.

**Gene knockdown**. Knockdown of STT3A, STT3B, TOP2A, and TOP2B was performed using CRISPR/Cas9 and HDR plasmids (Santa Cruz) according to the manufacturer's instructions. Briefly, the cells were co-transfected with CRISPR/Cas9 and HDR plasmids. After 2 days, the cells were selected with puromycin for 1 week and subjected to analyses. A control CRISPR/Cas9 plasmid encoding a non-specific 20 nt guide RNA (Santa Cruz) was used as a negative control. β-catenin (CTNNB1) knockdown was performed by siRNA (Sigma). The target sequences are listed in Supplementary Table 2.

**Flow cytometry analysis**. To perform flow cytometry analysis, single-cell suspensions were prepared and resuspended in stain buffer (BD Biosciences). Human tumor cells were stained according to standard protocols for flow cytometry with the following antibodies: BV421-CD24 (BD Biosciences, #562789, 1:100), FITC-CD44 (BD Biosciences, #555478, 1:100), and/or APC-PD-L1 (BioLegend, #329708, 1:50). Mouse tumor cells were stained with the following antibodies: APC-CD24 (BD Biosciences, #562349, 1:100), APC-CD133 (BioLegend, #141207, 1:100), PE-CD44 (BD Biosciences, #553134, 1:100), BV421-PD-L1 (BioLegend, #124315, 1:100), PE/Cy7-E-cadherin (BioLegend, #147309, 1:100), and/or Alexa Fluor 488-N-cadherin (R & D Systems, #FAB6426G, 1:50). To further intracellular ALDH1A1, cells were fixed, permeabilized, and stained with anti-ALDH1A1 (abcam, #ab52492, 1:1000). The cells were washed twice and then stained with secondary antibodies conjugated with Alexa Fluor 488 (Life Technologies). Isotype IgG or secondary antibody alone were used as negative control. Human breast CSCs were isolated by sorting for CD44$^+$CD24$^{-/low}$ cells[21]. 4T1 mouse breast CSCs were isolated by sorting for CD44$^+$CD24$^+$ALDH1$^+$ cells[22]. CT26 mouse colon CSCs were isolated by sorting for CD44$^+$CD133$^+$ALDH1$^+$ cells[61]. TILs were stained with the following antibodies: PerCP-CD3 (BioLegend, #100326, 1:100), PE-CD45 (BioLegend, #103105, 1:100), APC/Cy7-CD8a (BioLegend, #100713, 1:100), then fixed, permeabilized, and stained with Pacific blue-IFNγ (BioLegend, #505817, 1:50). Stained samples were evaluated by BD FACS Canto II (BD Immunocytometry Systems) and analyzed by FlowJo.

**PD-1 binding assay**. To analyze the PD-1 binding ability of cells, single-cell suspensions ($1 × 10^6$ cells) were incubated with 5 μg ml$^{-1}$ of recombinant human PD-1 Fc chimera protein (for human cells) or recombinant mouse PD-1 chimera protein (for 4T1 cells) (R & D Systems) at room temperature for 30 min. After washing, the cells were stained with fluorescence-conjugated anti-human IgG secondary antibody. Secondary antibody alone were used as negative control. After staining, the cells can be subjected to analysis or further immunostaining of other molecules according to standard protocols for flow cytometry. The immunofluorescence was evaluated by BD FACS Canto II (BD Immunocytometry Systems) and analyzed by FlowJo.

**In vitro PBMC-mediated cancer cell killing assay**. PBMC-mediated cancer cell killing assay was modified from previously described methods[62,63]. To activate PBMC, $1 × 10^7$ PBMC cells (STEMCELL Technologies and iQ Biosciences) were incubated with $1 × 10^6$ targeting cancer cells (BT-549 or 4T1 cells) in the presence of anti-CD3 antibody (Life Technologies, 100 ng ml$^{-1}$) and IL-2 (10 ng ml$^{-1}$) for 4 days. After incubation, activated PBMCs were purified by Percoll (GE Healthcare) density gradient.

To assay PBMC-mediated cancer cell killing of the CSC and non-CSC populations of BT-549 cells in vitro, BT-549 cells were labeled with CellTrace (Far Red, Invitrogen) according to the manufacturer's instructions. After labeling, $0.5 × 10^6$ BT-549 cells were mixed with $0.5 × 10^7$ BT-549-primed human PBMCs in 0.1 ml medium containing human CD3/CD28 tetrameric antibody complexes (STEMCELL Technologies) in polystyrene tube (FALCON). The BT-549/PBMC cell mixture was incubated at 37 °C, 5% CO$_2$ in a humidified incubator for 48 h. After incubation, the cells were washed with PBS and fixed and permeabilized with Fix/Perm solution (BD Biosciences) according to the manufacturer's instructions. The cells were then stained with the following antibodies: PE-CD44 (BD Biosciences, #555479, 1:50), FITC-CD24 (BD Biosciences, #555427, 1:50), and V450-cleaved caspase 3 (BD Biosciences, #560627, 1:50) and analyzed by BD FACS Canto II (BD Immunocytometry Systems). The BT-549 cell populations were first isolated from the BT-549/PBMC cell mixture by gating for the CellTrace$^+$ population. Then, the CSC and non-CSC populations of BT-549 cells were isolated by gating for CD44$^+$CD24$^{-/low}$ and non-CD44$^+$CD24$^{-/low}$ populations. The cytotoxicity status of CSC and non-CSC populations were analyzed through detection of cleaved caspase 3 signal.

To assay PBMC-mediated cancer cell killing of 4T1 sphere cells in vitro, single-cell suspensions of 4T1 spheres were prepared and labeled with CellTrace. After labeling, $0.5 × 10^6$ sphere cells were mixed with $0.5 × 10^7$ 4T1-primed BALB/c mouse PBMCs in 0.1 ml medium containing Dynabeads mouse T-activator CD3/CD28 (Invitrogen) in polystyrene tube. After 48-h incubation, the 4T1 sphere/PBMC mixture was harvested, permeabilized, and stained with V450-cleaved caspase 3 (BD Biosciences, #560627, 1:50). The 4T1 sphere cells were isolated by gating for the CellTrace$^+$ population and the cytotoxicity status of 4T1 sphere cells was analyzed through detection of cleaved caspase 3 signal.

**Duolink assay (in situ proximity ligation assay)**. Cells were seeded in chamber slides (Nunc Lab-Tek). To harvest the cells, cells were washed with cold PBS twice and fixed with 4% paraformaldehyde at 4 °C for 2 h. After PBS washing, cells were permeabilized by cold 0.2% Triton X-100 (Sigma) for 30 min at room temperature and subjected to Duolink assay (Olink Bioscience) according to the manufacturer's instructions. The positive signal is visualized as distinct fluorescence spot and each spot represents one cluster of protein–protein interaction. Antibodies used for Duolink assay were as follows: non-phospho (active) β-catenin (Cell Signaling, #8814, 1:500) and TOP2B (R&D Systems, #MAB6348, 1:100).

**Immunohistochemistry staining**. Breast cancer tissue microarrays ($n = 129$) were collected from Shanghai Jiaotong University, and written informed consent was obtained from patients in all cases at the time of enrollment. Tissues specimens were incubated with antibodies against E-cadherin (BD Biosciences, #610182, 1:100), STT3A (Santa Cruz, #SC-100796, 1:50), STT3B (abcam, #ab122351, 1:150), non-phospho (active) β-catenin (Cell Signaling, #8814, 1:100), or PD-L1 (abcam, #ab205921, 1:100) at 4 °C overnight. The specimens were then treated with biotinylated secondary antibody, followed by incubations with

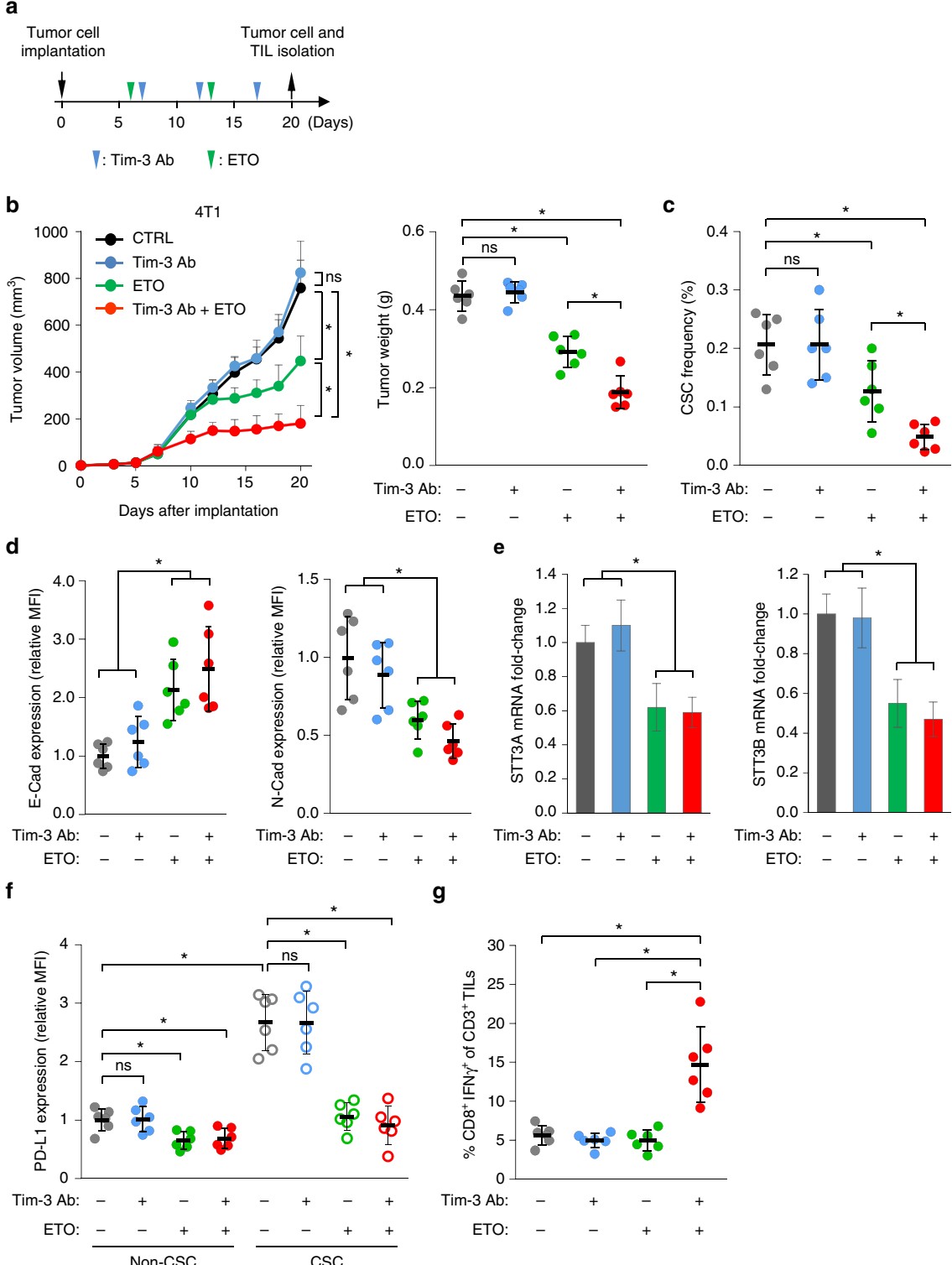

**Fig. 7** Etoposide enhances the therapeutic efficacy of Tim-3 blockade therapy. **a** Schematic diagram illustrating the treatment protocol of Tim-3 antibody (Tim-3 Ab) and/or etoposide (ETO) in mice. At the endpoint, tumor cells and tumor-infiltrating lymphocytes (TIL) were isolated for analysis. **b** Tumor growth of 4T1 cells in BALB/c mice treated with Tim-3 antibody and/or etoposide. Tumor size was measured at the indicated time points and tumor weight was measured at the endpoint ($n = 6$ mice per group). **c** Efficacy of indicated treatments on the CSC (CD44$^+$CD24$^+$ALDH1$^+$ populations) frequency of 4T1 tumors. **d** Flow cytometric analysis analyzing the effect of indicated treatments on the expression of E-cadherin (E-Cad, epithelial marker) and N-cadherin (N-Cad, mesenchymal marker) in the entire cancer cell population of 4T1 tumors. **e** qRT-PCR analysis of the influence of indicated treatments on STT3 isoforms expression in the entire cancer cell population of 4T1 tumors. **f** Effect of indicated treatments on PD-L1 expression in the CSC and non-CSC populations of 4T1 tumors. **g** Intracellular cytokine staining of CD8$^+$ IFNγ$^+$ cells in the CD3$^+$ T cell populations from isolated tumor-infiltrating lymphocytes to analyze the impacts of indicated treatments on tumor-infiltrating cytotoxic T cell activity. Error bars represent s.d. ($n = 6$). *$P < 0.05$; ns: non-significant, Student's $t$-test. See also Supplementary Fig. 7

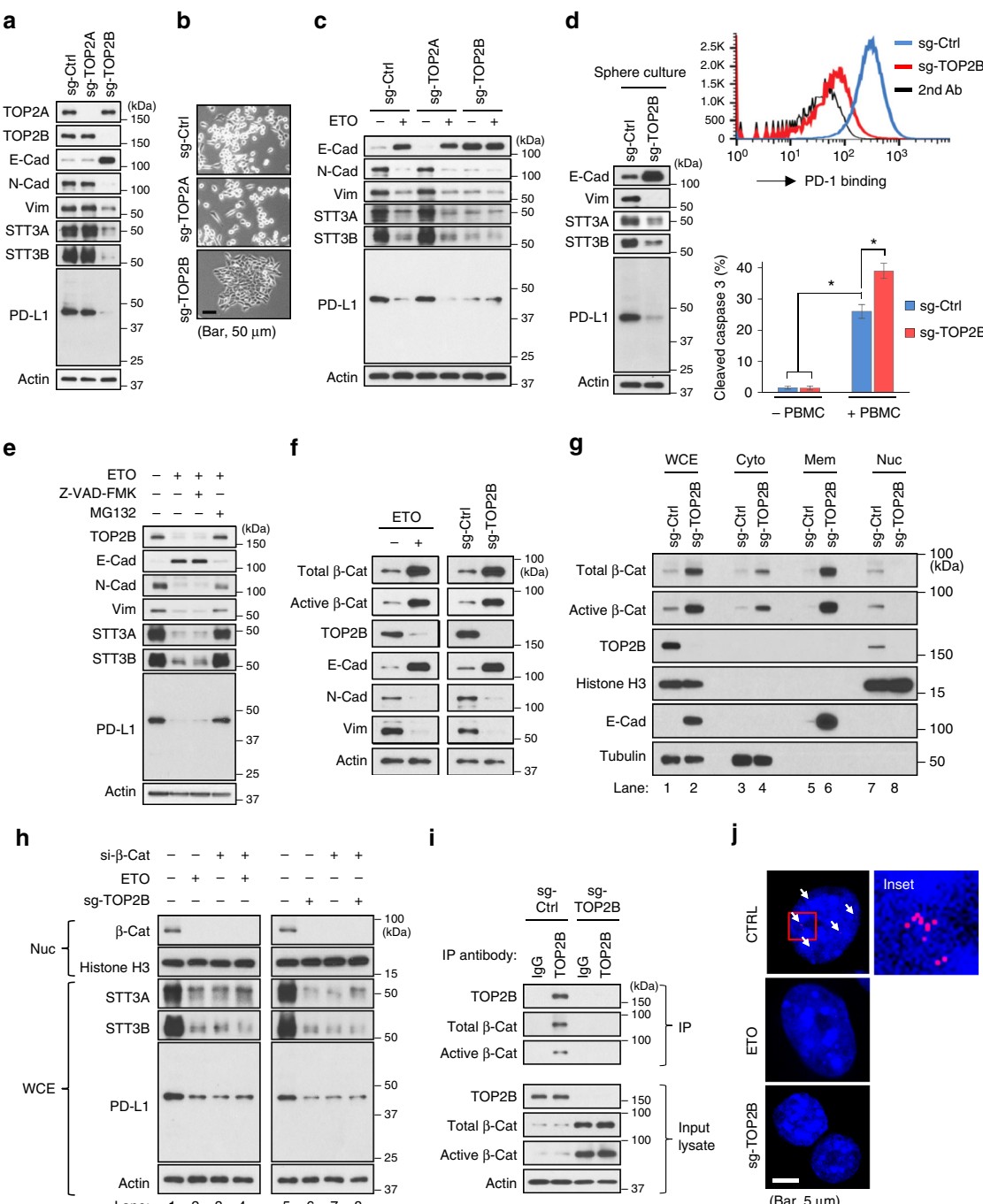

**Fig. 8** Etoposide inhibits the EMT/β-catenin/STT3/PD-L1 axis through TOP2B degradation-dependent nuclear β-catenin downregulation. **a** Effect of TOP2 isoforms knockdown on the expression of EMT markers, STT3 and PD-L1 in 4T1 cell. **b** Representative phase-contrast microscopy images of 4T1 cell treated with indicated sg-RNAs. Scale bar, 50 μm. **c** Influence of TOP2 isoforms knockdown on etoposide-induced MET and downregulation of STT3 and PD-L1. **d** Western blot analysis (left), PD-1 binding assay (upper right), and in vitro PBMC-mediated cancer cell killing assay (bottom right) of tumorspheres cultured from various sgRNA-treated 4T1 cells. **e** Effect of proteasome inhibitor (MG132) and caspase inhibitor (Z-VAD-FMK) on etoposide-induced TOP2B degradation, MET, and downregulation of STT3 and PD-L1. **f** Western blotting of whole-cell extracts analyzing the effect of etoposide and TOP2B knockdown on total and active (non-phospho) β-catenin (β-Cat). **g** Western blotting of whole-cell extracts (WCE), cytosolic (Cyto), membrane (Mem), or nuclear (Nuc) fractions from sgRNA-treated 4T1 cells analyzing the effect of TOP2B knockdown on β-catenin subcellular localizations. Histone H3, E-cadherin (E-Cad), and tubulin were used as makers of nuclear, membrane, and cytosolic fractions, respectively. **h** Efficacy of β-catenin knockdown (si-β-Cat) in desensitizing cells to etoposide- and sg-TOP2B- induced MET and downregulation of STT3 and PD-L1. Nuc nuclear fraction, WCE whole-cell extracts. **i** Co-immunoprecipitation assay showing interaction between TOP2B and β-catenin. **j** Duolink assay analyzing the interaction of TOP2B and active (non-phospho) β-catenin in the nucleus. Red dots (indicated by arrows) represented TOP2B-β-catenin interaction signals and nuclei were counterstained with DAPI (blue). The right column shows higher-magnification image of the area outlined in the left column. Scale bar, 5 μm. Error bars represent s.d. (n = 3). *P < 0.05, Student's t-test. See also Supplementary Fig. 8

avidin–biotin–peroxidase complex solution for 1 h at room temperature. Visualization was performed using 3-amino-9-ethylcarbazole solution. Counterstaining was carried out using Mayer's hematoxylin. All immunostained slides were scanned on the Automated Cellular Image System III (ACIS III, Dako, Denmark) for quantification by digital image analysis. The score of protein expression was calculated from both the percentage (0–100%) of immunopositive cells and the immunostaining intensity (0: negative, 1: low, 2: medium, 3: high). The number from percentage × intensity represents an arbitrary quantitative score[64]. According to the scores, the protein expression correlations between different proteins were analyzed.

**ConA lectin binding assay.** Immunopurified PD-L1 proteins were subjected to SDS-PAGE, transferred onto PVDF (Bio-Rad), and detected by peroxidase-conjugated ConA lectin (Sigma) according to the manufacturer's instructions.

**Luciferase reporter assay.** The putative promoter regions of STT3A ($-1042$ to $+210$) and STT3B ($-1381$ to $-1$) were cloned and fused to a Gaussia luciferase gene. MCF10A cells were transfected with promoter-luciferase reporter constructs combined with CMV/red firefly vector as an internal standard. After incubation for 48–72 h, the cells were rinsed with PBS and subjected to luciferase assay using a luciferase dual assay system (Pierce) according to the manufacturer's instruction.

**CCLE and TCGA gene expression datasets and data processing.** The mRNA expression (RNA Seq V2 RSEM) database of TCGA Breast Invasive Carcinoma ($n = 1100$) was downloaded from the open-source resource of the cBioPortal for Cancer Genomics [http://www.cbioportal.org/index.do]. The log2 ratio of downloaded data was determined using Cluster 3.0 software [http://bonsai.hgc.jp/~mdehoon/software/cluster/software.htm]. The EMT scores of each samples were calculated according to mRNA expression signatures of 37 core EMT-related genes[31], comprising 20 mesenchymal genes and 17 epithelial genes, by determining the mean log ratio of the genes in the "mesenchymal" arm of the signature and then subtracting the mean log ratio of the genes in the "epithelial" arm (Supplementary Data 1). A higher EMT score indicates a more mesenchymal-like signature. To view the clustering results generated by Cluster 3.0, we use Alok Saldanha's Java TreeView [http://sourceforge.net/projects/jtreeview/], which can display both hierarchical and k-means clustering results. The heat map was represented graphically by coloring each samples on the basis of the measured fluorescence ratio. For mRNA expression, log ratios of 0 (a ratio of 1.0 indicates that the genes are unchanged) were colored in black, positive log ratios were colored in red, and negative log ratios were colored in green (with darker colors corresponding to higher ratios). For EMT score, the score of 0 was colored in gray, positive scores were colored in yellow, and negative scores were colored in blue (with darker yellow corresponding to more mesenchymal and darker blue corresponding to more epithelial).

**Cell mobility, migration, and invasion assays.** Time-lapse microscopic observations of cell mobility were performed in a humidified, $CO_2$-equilibrated chamber with an Axiovert 200 M cell observer microscope (Carl Zeiss). The cells were pre-treated with chemical compounds for 16 h and then were observed for 6 h. Images were obtained with a high-resolution digital charge-coupled device camera (AxioCam HRm, Carl Zeiss) and analyzed using the AxioVision 4.8 imaging software (Carl Zeiss). The mobility of 20 randomly selected cells were tracked and the mobility speeds were presented as μm min$^{-1}$. Cell migration and invasion assays were performed using Biocoat Control inserts and Biocoat Matrigel invasion chambers (Corning), respectively. The cells were pre-treated with chemical compounds for 16 h and then dissociated with trypsin, washed, and resuspended as single-cell suspensions in DMEM medium with 0.1% FBS. Single-cell suspensions were plated to the upper chamber and allowed to penetrate a porous (8 μm), uncoated membrane or a Matrigel-coated membrane to the bottom chamber containing DMEM medium with 10% FBS. Cells on the upper side of the membrane were removed after 4-h and 12-h incubation, respectively. The cells on the underside were fixed with 4% paraformaldehyde, stained with 0.5% crystal violet, and counted from four randomly selected fields of each membrane. The average cell number per field for each membrane was used to calculate the mean and s.d. for triplicate membranes.

**Quantitative real-time PCR (qRT-PCR).** Total RNA was extracted using RNeasy Plus Mini Kit (QIAGEN) according to the manufacturer's instructions and then subjected to complementary DNA by reverse transcription using the SuperScript III kit (Invitrogen). PCR reactions were performed in triplicate with iQ SYBR Green Supermix (Bio-Rad) in the iCycler iQ system (Bio-Rad). The mRNA levels of target genes were normalized to 18S rRNA using QuantumRNA 18S Internal Standards (Life Technologies). Primer pairs used for qRT-PCR are listed in Supplementary Table 2.

**Animal studies, treatment, and tumor tissue staining.** All animal procedures were conducted under the guidelines approved by the Institutional Animal Care and Use Committee (IACUC) at MD Anderson Cancer Center. To study the effect

of etoposide on tumorsphere-mediated tumorigenesis, sphere cells cultured from 4T1 cells in vitro were dissociated, washed, resuspended in PBS as single-cell suspensions, and cell viability was analyzed by trypan blue exclusion. Serial dilutions of viable sphere cells were implanted into the mammary fat pads of syngeneic 6-week female BALB/c mice (Jackson Laboratories). Tumor initiation and growth were monitored for 1 month after cell injection. Tumor volumes were measured with a caliper and determined using the formula $l \times w \times w/2$, where $l$ is the longest diameter and $w$ is the shortest diameter. To perform tumor tissue staining, comparably sized tumors from each group were used. Under anesthesia, mice were perfused with 0.1 M phosphate-buffered saline (PBS; pH 7.4) and tumor masses were frozen immediately after extraction. Cryostat sections (5-μm thick) were fixed with 4% paraformaldehyde for 15 min at room temperature. Fixed cryostat sections were blocked with blocking solution (1% BSA, 0.5% Triton X-100, 2% donkey serum, 0.01 M PBS, pH 7.2) at room temperature for 30 min. Samples were stained with primary antibodies overnight at 4 °C, followed by Alexa 488, 549, and 647 (Invitrogen) secondary antibodies at room temperature for 1 h. Nuclear staining was performed with Hoechst 33342 (Molecular Probes). Images were obtained using confocal microscope (Carl Zeiss, LSM700) and analyzed by ImageJ. Antibodies used for tissue staining were as follows: PD-L1 (Cell Signaling, #64988, 1:200), CD8 (marker for cytotoxic T lymphocyte; abcam, #ab22378, 1:200), and granzyme b (marker for activated cytotoxic T lymphocyte; R&D Systems, #AF1865, 1:100).

To study the therapeutic effect of etoposide and/or Tim-3 antibody in preclinical tumor models, 4T1 ($5 \times 10^4$ cells) or CT26 cells ($5 \times 10^5$ cells) were suspended in 50 μl of medium mixed with 50 μl of Matrigel Basement Membrane Matrix (BD Biosciences) and injected subcutaneously into 6-week female BALB/c mice (Jackson Laboratories). For Tim-3 antibody treatment, 100 μg of Tim-3 antibody (clone B8.2C12; Bio X Cell) or control rat IgG (clone HRPN; Bio X Cell) was injected intraperitoneally on days 7, 12, and 17 after tumor cell inoculation. For etoposide treatment, mice were treated intravenously with 50 mg kg$^{-1}$ etoposide (LC Laboratories; prepared in buffer containing 2 mg ml$^{-1}$ citric acid, 30 mg ml$^{-1}$ benzyl alcohol, 80 mg ml$^{-1}$ polysorbate 80, 650 mg ml$^{-1}$ polyethylene glycol 300, and 30.5% alcohol) on days 6 and 13 after tumor cell inoculation. Tumor volumes were measured every 2 days with a caliper and calculated using the formula $l \times w \times w/2$, where $l$ is the longest diameter and $w$ is the shortest diameter. At day 20, tumor weight was measured and single-cell suspensions of tumor cells and TILs were prepared using tumor dissociation kit (Miltenyi Biotec) and gentleMACS dissociator (Miltenyi Biotec) followed by Percoll (GE Healthcare) density gradient enrichment, and then subjected to flow cytometric analysis.

**Quantification of N-glycan site occupancy (%) of PD-L1.** To quantify the N-glycan occupancy (%) of each PD-L1 N-glycosylation site before and after EMT induction, total PD-L1 proteins, including glycosylated and non-glycosylated forms, were purified from epithelial and mesenchymal cells individually. To this end, Flag-PD-L1 was expressed in epithelial (untreated MCF-10A) or mesenchymal (TGF-β-treated MCF-10A) cells in the presence of proteasome inhibitor MG132 (to prevent non-glycosylated PD-L1 from degradation). Then, both glycosylated and non-glycosylated Flag-PD-L1 proteins were purified by Flag-beads and subjected to LC-MS/MS analysis. The data were performed on an Orbitrap Fusion Lumos Tribrid mass spectrometer (Thermo Scientific), fitted with a PicoView nanospray interface (New Objective) and Easy-nLC 1200 (Thermo Scientific). N-glycan occupancy (%) of each N-glycosylation site was estimated based on the relative peak intensities of non-N-glycosylated and de-N-glycosylated peptides detected after removal of N-glycans by PNGase F. Enzymatic release of N-glycans would convert the originally occupied Asn (N) into Asp (D), creating a one mass unit difference from the corresponding peptide carrying a non-occupied site, which could be resolved by extracted ion chromatograms at 10 ppm accuracy based on the calculated accurate mass. To account for spontaneous deamidation as opposed to true enzymatic de-N-glycosylation, a control experiment was performed in which PNGase F was omitted. The intensities of two non-glycosylated tryptic peptides from PDL1, DQLSLGNAALQITDVK ($m/z$ 843.4570) and AEVIWTSSDHQVLSGK ($m/z$ 586.3003), were used to normalize the protein amount among samples. Assuming the recovery and MS response of a tryptic peptide carrying the N×T site relative to the same peptide carrying the de-N-glycosylated or deamidated D×T site is approximately the same, the % site occupancy can be calculated based on the formula (INF−IND) / (IN + INF) × 100, in which INF = normalized intensity of peptide with D×T, after PNGase F treatment; IND = normalized intensity of peptide with D×T from the sample without PNGase F treatment (spontaneous deamidation); IN = normalized intensity of peptide with N×T from the sample treated with PNGase F. A 100% occupancy would imply IN = 0, with no spontaneous deamidation (IND = 0).

**Statistical analyses.** Statistical analyses were performed with Microsoft Excel analysis tools or GraphPad Prism software. All data were presented as means ± standard deviation (s.d.). Student's $t$-test was used to compare two groups ($P < 0.05$ being considered statistically significant).

**Data availability.** All data supporting the findings of this study are available with the article, or from the corresponding author upon reasonable request.

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

## Acknowledgements

This work was funded in part by the following: National Institutes of Health (CCSG CA016672); Cancer Prevention & Research Institutes of Texas (DP150052 and RP160710); National Breast Cancer Foundation, Inc.; Breast Cancer Research Foundation (to M.C.H. and G.N.H.); Patel Memorial Breast Cancer Endowment Fund; The University of Texas MD Anderson-China Medical University and Hospital Sister Institution Fund; Ministry of Science and Technology, International Research-intensive Centers of Excellence in Taiwan (I-RiCE; MOST 105-2911-I-002-302); Ministry of Health and Welfare, China Medical University Hospital Cancer Research Center of Excellence (MOHW106-TDU-B-212-144003); Center for Biological Pathways; the National Research Foundation of Korea grant for the Global Core Research Center funded by the Korea government (MSIP) (2011-0030001 to J.-H.C.). We also would like to acknowledge the Academia Sinica Common Mass Spectrometry Facilities located at the Institute of Biological Chemistry for mass spectrometry data acquisition.

## Author contributions

J.-M.H. designed and performed the experiments, analyzed the data, and wrote the manuscript; Y.-H.H., L.-C.C., W.-H.Y., J.-H.C., C.-T.C., H.-W.L. C.-W. K., and K.-H. K. performed the experiments and analyzed the data; W.X. and G.-X.R. provided the patient tissue samples and performed the immunohistochemical staining; J.L.H., C.-W.L., S.-O. L., S.-S.C., and Y.-C.C. provided scientific input and wrote the manuscript; M.-C.H. supervised the entire project, designed the experiments, analyzed the data, and wrote the manuscript.

## Additional information

**Competing interests:** The authors declare no competing interests.

