## [Peer Review File · Nature Communications]

Reviewers' comments:

Reviewer #1 (Remarks to the Author):

In the manuscript entitled "Mesenchymal-epithelial transition by etoposide inhibits PD-L1 glycosylation and enhances vulnerability to antitumor immunity", the authors demonstrated that EMT stabilizes PD-L1 by the EMT- β -catenin-STT3-PD-L1 axis. Mechanistically, EMT transcriptionally induces N-glycosyltransferase STT3A/B through β -catenin/TCF4, and subsequent STT3-dependent PD-L1 N-glycosylation stabilizes and upregulates PD-L1. They further found that etoposide could promote PD-L1 degradation through TOP2B degradation-dependent nuclear β -catenin reduction and sensitize cancer cells to anti-Tim-3 therapy. This manuscript describes interesting findings on the link between EMT and PD-L1 stabilization, and is of great clinical importance as a potential strategy to enhance cancer immunotherapy efficacy, it is thus recommended for publication if the following issues can be addressed.

1. The authors need to provide the guide RNA sequences they used to knockdown genes.
2. In Supplementary Fig 1a-c, what were the CD274 mRNA levels of non-CSC and CSC population cells?
3. MCF10A literally is not a cancer cell line, so it's not very proper to describe CD44+CD24-/low MCF10A as CSC.
4. In Supplementary Fig 3b, what's the mRNA levels of STT3A/B and CD274 (PD-L1)?
5. In Fig 3c, will stimulation of TGF- β further upregulate STT3A/B in BT-549 CSC population?
6. In Fig 4c, including STT3A/B reporters with mutated TCF4 binding sites would be ideal to strengthen authors' conclusion.
7. It would be ideal if the authors could show the STT3A/B and PD-L1 modulation effects of β -Catenin knockdown in mesenchymal-like TNBC cells (Fig 4).
8. In Fig 6, rescue assays with ectopic expression of STT3A/B would further strengthen authors' conclusion.

Reviewer #2 (Remarks to the Author):

This manuscript attempts to build a story by which increase of STT3 expression by EMT selectively causes increases in PD-L1 expression and altered cytotoxicity by PBMC. In addition, etoposide is seen to affect this pathway and exert its effects on PD-L1. There are eight figures and extensive supplementary figures describing experiments using cancer stem cell and non-stem cell populations, PDL-L1, beta-catenin regulatory effects, and, finally, etoposide effects. There are at least two manuscripts here, especially if extensive controls are performed and shown. Unfortunately, the fundamental premise, that EMT upregulates STT3 isoforms to cause stabilization and increased PDL-1 glycoprotein, although interesting, is not demonstrated unambiguously. The beta-catenin and etoposide experiments that follow simply extend this premise.

1. The study concentrates on STT3 isoforms, an essential component of the complex that N-glycosylates proteins with the Asn-X-Ser motif and proper folding. Many glycosyltransferases show an increase in EMT; for example, ST6Gal1. No discussion is ever made about more distal glycosylation having an effect on PDL-1 levels of expression and its ability to be bound by PD-1. When STT3 levels are altered experimentally (knock down/out and over-expression, it is predicted

that thousands of glycoproteins will be altered in their expression levels. Likewise, treatment of cells with tunicamycin affects thousands of glycoproteins by decreasing their N-glycosylation. Its effects may be consistent with the hypothesis, but in this case an association does not imply causality—the effects on PDL-1 could be caused by other mechanisms affected by tunicamycin treatment. Controls should include effects on expression of other glycoproteins with multiple N-linked sites.

2. In essence, the argument is that as endogenous STT3 is increased during EMT, and this is not particularly well quantified, it and it alone increases N-glycosylation of un-glycosylated PDL-1, which has four potential N-linked sites that appear to be utilized. This increased N-glycosylation keeps a population of PDL-1 from being mis-folded and degraded, leading to more PDL-1 on the cell surface. The authors have done some site-mapping in a previous publication—this could be extended with proteomics to determine the % occupancy at each site before and after induction of EMT. With four sites, it is not at all clear what the requirements for “escape” from degradation are in terms of site occupancy. There are probably significant differences among different breast cancer cell lines and possibly differences between cancer stem cells and cancer non-stem cells. Con A binds to high-mannose, hybrid, and bi-antennary glycans and not to branched glycans, which would be predicted to be increased after EMT due to induction of specific glycosyltransferases. LC MS/MS analysis should demonstrate what is actually resulting from the EMT inducers.

3. In Line 188, it is not clear what, “enriches PDL1-expression of CSCs” means.

4. The effects of etoposide are interesting, but these effects should be the subject of a separate manuscript, after extensive documentation of the effects of the increases of endogenous STT3 on PDL-1 as well as control, cell surface glycoproteins.

In sum, the conclusions of the manuscript regarding EMT-mediated increased STT3-mediated glycosylation causing selective increases in PDL-1 expression on the cell surface require additional experimentation to be substantiated.

Reviewer #3 (Remarks to the Author):

This is an interesting study showing that EMT triggers N-glycosylation of PD-L1, leading to enriched PD-L1 in CSCs via the b-catenin/STT3 axis. Previous studies by the same group indicate that the glycosylation of PD-L1 could stabilize PD-L1 expression on tumor cells and increase resistance to immune attack. These findings may help understand the mechanisms of CSCs in immune evasion. The paper is well-written and the conclusions are supported by the results in general.

Specific comments:

1. As we learn from numerous clinical studies, humanized anti-PD-1 or anti-PD-L1 antibodies can well saturate these molecules in vivo and provide occupancy of these targets at least a few weeks. The results presented in this paper may indicate a different way to down-regulate the expression of PD-L1 on cancer cells, including CSCs. While etoposide study presented here is interesting, this drug did not achieve the same efficacy as well as durability in cancer therapy as anti-PD-1/PD-L1 antibodies in clinic. Therefore, etoposide may only partially or less effectively modulate the expression of PD-L1 as its mechanism of action. In this context, the authors should carefully state the significance of their findings and discuss these possibilities.

2. The assays for cytotoxic T cells with the stimulation of human PBMCs activation in vitro by xenogeneic antigens and anti-CD3/IL-2 is highly artificial. While this assay indicates the change of cytotoxic activity of T cells, its significance is limited because this activity is much higher and well beyond normal T cell activity in real life. Therefore, I suggest that the authors to state the fact and do not use this data to implicate antitumor activity in vivo.

Point-by-Point Response to Reviewer's Comments

Reviewer #1 (Remarks to the Author):

In the manuscript entitled “Mesenchymal-epithelial transition by etoposide inhibits PD-L1 glycosylation and enhances vulnerability to antitumor immunity”, the authors demonstrated that EMT stabilizes PD-L1 by the EMT- β -catenin-STT3-PD-L1 axis. Mechanistically, EMT transcriptionally induces N-glycosyltransferase STT3A/B through β -catenin/TCF4, and subsequent STT3-dependent PD-L1 N-glycosylation stabilizes and upregulates PD-L1. They further found that etoposide could promote PD-L1 degradation through TOP2B degradation-dependent nuclear β -catenin reduction and sensitize cancer cells to anti-Tim-3 therapy. This manuscript describes interesting findings on the link between EMT and PD-L1 stabilization, and is of great clinical importance as a potential strategy to enhance cancer immunotherapy efficacy, it is thus recommended for publication if the following issues can be addressed.

- 1. The authors need to provide the guide RNA sequences they used to knockdown genes.*
- 2. In Supplementary Fig 1a-c, what were the CD274 mRNA levels of non-CSC and CSC population cells?*
- 3. MCF10A literally is not a cancer cell line, so it's not very proper to describe CD44+CD24-/low MCF10A as CSC.*
- 4. In Supplementary Fig 3b, what's the mRNA levels of STT3A/B and CD274 (PD-L1)?*
- 5. In Fig 3c, will stimulation of TGF- β further upregulate STT3A/B in BT-549 CSC population?*
- 6. In Fig 4c, including STT3A/B reporters with mutated TCF4 binding sites would be ideal to strengthen authors' conclusion.*
- 7. It would be ideal if the authors could show the STT3A/B and PD-L1 modulation effects of β -Catenin knockdown in mesenchymal-like TNBC cells (Fig 4).*
- 8. In Fig 6, rescue assays with ectopic expression of STT3A/B would further strengthen authors' conclusion.*

Authors' Response to Reviewer #1's Comments:

We deeply appreciate the reviewer for the comments and suggestions to improve the scientific merit of the manuscript. Below please find our response to each of these comments.

Point #1: *The authors need to provide the guide RNA sequences they used to knockdown genes.*

Response: We sincerely appreciate the reviewer's constructive comments. The target sequences are shown below and have been incorporated in the **Methods** section of revised manuscript (**page 24**).

	Target sequence
Human STT3A sgRNA (Santa Cruz, sc-405155)	ACAGACATTCCGAATGTCGA AAGGTGGTACGTGACGATGG CTCGGTCATCAAACCAGTTA
Human STT3B sgRNA (Santa Cruz, sc-404481)	GATGTAAGGCCGCTAAAAGT CCAGCGGTTATCATCAACCC TACAGCAAAAGAGTCTACAT
Human β -catenin siRNA (Sigma, EHU139421)	GCCGGCTATTGTAGAAGCTGGTGGGA ATGCAAGCTTTAGGACTTCACCTGA CAGATCCAAGTCAACGTCTTGTTC GAACTGTCTTTGGACTCTCAGGAAT CTTTCAGATGCTGCAACTAACAGG AAGGGATGGAAGGTCCTTTGGGAC TCTTGTTCAGCTTCTGGGTTGAGAT GATATAAATGTGGTCACCTGTGCAG CTGGAATTCTTTCTAACCTCACTTG CAATAATTATAAGAACAAGATGATG GTCTGCCAAGTGGGTGGTATAGAGG CTCTGTGCGTACTGTCTTCCGGGC TGGTGACAGGGAAGACATCACTGAG CCTGCCATCTGTGCTCTTCGTGATC TGACCAGCCGACACCAAGAAGCAGA GATGGCCCAGAATGCAGTTCGCCTT CACTATGGACTACCAGTTGTGGTTA AGCTCTTACACCCACCATCCCACT
Mouse β -catenin siRNA (Sigma, EMU047621)	AGGGTGGGAATGGTTTATAGCCCTGT TTGTAAATCTGCCACCAAACAGATA CATACTTGGGAAGGAGATGTTTCATG TGTGGAAGTTTCTCACGTTGATGTT TTTGCCACAGCTTTTGCAGCGTTAT ACTCAGATGAGTAACATTTGCTGTT TTCAACATTAATAGCAGCCTTTCTC TCTATACAGCTGTAGTGTCTGAACG TGCATTGTGATTGGCCTGTAGAGTT GCTGAGAGGGCTCGAGGGGTGGGCT GGTATCTCAGAAAGTGCCTGACACA CTAACCAAGCTGAGTTTCTATGGG AACAGTCGAAGTACGCTTTTGTTC TGGTCTTTTGGTTCGAGGAGTAAC AATACAAATGGATTGGGGAGTGAC TCACGCAGTGAAGAATGCACACGAA TGGATCACAAG
Mouse TOP2A sgRNA (Santa Cruz, sc-423469)	CTCATCGTCGTCATAGTTAC CTCCGCCCAGATACCTACAT TGGGTTTACGATGAAGATGT
Mouse TOP2B sgRNA (Santa Cruz, sc-423470)	CTTCGTCTGATACATACAT ACTGATCCAATGTATGTATC TCTACTTTGTGTTCTACTAC

Point #2: In Supplementary Fig 1a-c, what were the CD274 mRNA levels of non-CSC and CSC population cells?

Response: The CD274 mRNA levels of CSC and non-CSC populations of mesenchymal breast cancer cells BT-549, MDA-MB-231, and 4T1 do not show significant differences. The results have been incorporated into **Supplementary Figure 1**.

Point #3: MCF10A literally is not a cancer cell line, so it's not very proper to describe CD44+CD24-/low MCF10A as CSC.

Response: We apologize for inappropriately using CSC to describe the CD44+CD24-/low population of MCF10A cells. We modified our manuscript and described the CD44+CD24-/low population of MCF10A as "stem-like cell (SC) population" in **Figures 1** and **3** (shown below yellow highlight).

Fig 1

Fig 3

Point #4: In Supplementary Fig 3b, what's the mRNA levels of STT3A/B and CD274 (PD-L1)?

Response: Following the reviewer's comment, we carried out qRT-PCR analysis of STT3A, STT3B and CD274 (PD-L1) mRNA levels of the cell lines. Consistent with protein levels, STT3A/B and CD274 (PD-L1) mRNA levels were higher in low E-cadherin-expressing cells than in high E-cadherin-expressing cells. The results have been incorporated into **Supplementary Figure 4b** of the revised manuscript.

Point #5: In Fig 3c, will stimulation of TGF- β further upregulate STT3A/B in BT-549 CSC population?

Response: TGF- β did not significantly upregulate STT3A/B in the general cell population or CSC population of BT-549 cells (see figure below). This is likely because BT-549 is already a mesenchymal-like cell and therefore no longer responds to TGF- β -mediated EMT induction. The data are only shown in the rebuttal letter, but if reviewer feels this should be included in the revised manuscript, we would be happy to include it.

Point #6: In Fig 4c, including STT3A/B reporters with mutated TCF4 binding sites would be ideal to strengthen authors' conclusion.

Response: Following the reviewer's suggestion, we deleted the TCF4 binding sites within the STT3A/B promoter regions with results showing that STT3A/B reporters with mutated TCF4 binding site failed to be activated by β -Catenin, further supporting our notion that β -catenin/TCF4 transcriptionally activate both STT3 isoforms. The new results have been incorporated into **Figure 4c**.

Point #7: It would be ideal if the authors could show the STT3A/B and PD-L1 modulation effects of β -Catenin knockdown in mesenchymal-like TNBC cells (Fig 4).

Response: Following the reviewer's suggestion, we knocked down β -catenin in BT-549, a mesenchymal-like TNBC cell line, and showed that β -catenin knockdown downregulated STT3A/B and PD-L1. The new results have been incorporated into **Figure 4g**.

Point #8: In Fig 6, rescue assays with ectopic expression of STT3A/B would further strengthen authors' conclusion.

Response: Following the reviewer's suggestion, we ectopically expressed STT3A/B in etoposide-treated cells with results showing that exogenous STT3A/B rescued etoposide-induced PD-L1 downregulation. These data further supported the conclusion that etoposide downregulates PD-L1 through STT3A/B suppression. The new results have been incorporated into **Figures 6b, 6c** and **6e**.

Reviewer #2 (Remarks to the Author):

This manuscript attempts to build a story by which increase of STT3 expression by EMT selectively causes increases in PD-L1 expression and altered cytotoxicity by PBMC. In addition, etoposide is seen to affect this pathway and exert its effects on PD-L1. There are eight figures and extensive supplementary figures describing experiments using cancer stem cell and non-stem cell populations, PDL-L1, beta-catenin regulatory effects, and, finally, etoposide effects. There are at least two manuscripts here, especially if extensive controls are performed and shown. Unfortunately, the fundamental premise, that EMT upregulates STT3 isoforms to cause stabilization and increased PDL-1 glycoprotein, although interesting, is not demonstrated unambiguously. The beta-catenin and etoposide experiments that follow simply extend this premise.

1. The study concentrates on STT3 isoforms, an essential component of the complex that N-glycosylates proteins with the Asn-X-Ser motif and proper folding. Many glycosyltransferases show an increase in EMT; for example, ST6Gal1. No discussion is ever made about more distal glycosylation having an effect on PDL-1 levels of expression and its ability to be bound by PD-1. When STT3 levels are altered experimentally (knock down/out and over-expression, it is predicted that thousands of glycoproteins will be altered in their expression levels. Likewise, treatment of cells with tunicamycin affects thousands of glycoproteins by decreasing their N-glycosylation. Its effects may be consistent with the hypothesis, but in this case an association does not imply causality—the effects on PDL-1 could be caused by other mechanisms affected by tunicamycin treatment. Controls should include effects on expression of other glycoproteins with multiple N-linked sites.

2. In essence, the argument is that as endogenous STT3 is increased during EMT, and this is not particularly well quantified, it and it alone increases N-glycosylation of un-glycosylated PDL-1, which has four potential N-linked sites that appear to be utilized. This increased N-glycosylation keeps a population of PDL-1 from being mis-folded and degraded, leading to more PDL-1 on the cell surface. The authors have done some site-mapping in a previous publication—this could be extended with proteomics to determine the % occupancy at each site before and after induction of EMT. With four sites, it is not at all clear what the requirements for “escape” from degradation are in terms of site occupancy. There are probably significant differences among different breast cancer cell lines and possibly differences between cancer stem cells and cancer non-stem cells. Con A binds to high-mannose, hybrid, and bi-antennary glycans and not to branched glycans, which would be predicted to be increased after EMT due to induction of specific glycosyltransferases. LC MS/MS analysis should demonstrate what is actually resulting from the EMT inducers.

3. In Line 188, it is not clear what, “enriches PDL1-expression of CSCs” means.

4. The effects of etoposide are interesting, but these effects should be the subject of a separate manuscript, after extensive documentation of the effects of the increases of endogenous STT3 on PDL-1 as well as control, cell surface glycoproteins.

In sum, the conclusions of the manuscript regarding EMT-mediated increased STT3-mediated glycosylation causing selective increases in PDL-1 expression on the cell surface require additional experimentation to be substantiated.

Authors' Response to Reviewer's Comments:

We are deeply appreciative of the reviewer's comments and suggestions to improve the scientific merit of the manuscript. Below please find our point-by-point response to these comments.

Point #1-1: *The study concentrates on STT3 isoforms, an essential component of the complex that N-glycosylates proteins with the Asn-X-Ser motif and proper folding. Many glycosyltransferases show an increase in EMT; for example, ST6Gal1. No discussion is ever made about more distal glycosylation having an effect on PDL-1 levels of expression and its ability to be bound by PD-1.*

Response: We greatly appreciate the reviewer's detailed and constructive comments. Regarding the effects of distal glycosylation on PD-L1 expression levels and its ability to be bound by PD-1, we have not found any related published literature yet. However, we have a manuscript in press showing EGFR-induced β -1,3-Nacetylglucosaminyl transferase (B3GNT3) mediates the poly-LacNAc moiety on N192 and N200 glycosylation sites of PD-L1 and regulates PD-L1 binding with PD-1 (please see the figure below). This study implies that distal glycosylation is involved in the regulation of PD-L1/PD-1 binding by EGFR.

These data will be published in *Cancer Cell* soon and we have briefly discussed those results in the **Discussion** section of revised manuscript (**page 19, lines 8–12**) as follows:

“In addition to STT3-mediated incorporation of core glycan, our recent study also showed that β -1,3-Nacetylglucosaminyl transferase (B3GNT3)-mediated poly-LacNAc moiety on N192 and N200 glycosylation sites of PD-L1 is critical for PD-L1 binding with PD-1 (*Cancer Cell*, in press), suggesting that the distal portions of N-glycan on PD-L1 glycosylation sites are also functionally important in PD-L1 activity.”

Point #1-2: *When STT3 levels are altered experimentally (knock down/out and over-expression, it is predicted that thousands of glycoproteins will be altered in their expression levels. Likewise, treatment of cells with tunicamycin affects thousands of glycoproteins by decreasing their N-glycosylation. Its effects may be consistent with the hypothesis, but in this case an association does not imply causality—the effects on PDL-1 could be caused by other mechanisms affected by tunicamycin treatment. Controls should include effects on expression of other glycoproteins with multiple N-linked sites.*

Response: Following the reviewer’s suggestion, we incorporated tripeptidyl-peptidase 1 (TPP1) as a control of tunicamycin treatment. TPP1 is a lysosomal serine-carboxyl peptidase comprising five N-linked glycosylation sites. Previous studies have shown that N-glycosylation prominently regulates the protein stability of TPP1 [J Biol Chem 279, 12827-39 (2004)]. Our results indicated that tunicamycin downregulated TPP1 protein levels, which is consistent with the effects of tunicamycin on PD-L1 (see the figure below). However, even with TPP1 control, we felt that we still could not exclude the possibility that tunicamycin may affect PD-L1 protein levels through mechanisms other than PD-L1 glycosylation. In addition, the regulatory mechanism of tunicamycin on PD-L1 protein is not the main objective of our study. Furthermore, in the revised manuscript, we provided mass spectrometry analysis data to demonstrate the dynamic change of PD-L1 glycosylation upon EMT (see the response of Point #2), which supported the involvement of PD-L1 glycosylation in EMT-mediated PD-L1 induction. Therefore, **we removed the data of tunicamycin treatment (Supplementary Figure 2d-h) from our manuscript** without changing the major conclusion of the study.

The effects of tunicamycin on endogenous PD-L1 protein levels in mesenchymal-like breast cancer cell lines, MDA-MB-231 and 4T1. TPP1, whose protein stability is known to be regulated by N-glycosylation, was used as a positive control of tunicamycin treatment.

Regarding the effects of STT3 on PD-L1, in the Figure 2j and Supplementary Figure 4g of original manuscript (Figure 2j and Supplementary Figure 5g of revised manuscript), we showed that ectopic STT3A/B (Fig. 2j) or STT3A/3B knockdown (Supplementary Fig. 5g) affected the protein expression of wild-type (wt) PD-L1, but not unglycosylated PD-L1 mutant (4NQ), suggesting that STT3A/B regulates PD-L1 protein expression through PD-L1 glycosylation.

Point #2: *In essence, the argument is that as endogenous STT3 is increased during EMT, and this is not particularly well quantified, it and it alone increases N-glycosylation of un-glycosylated PDL-1, which has four potential N-linked sites that appear to be utilized. This increased N-glycosylation keeps a population of PDL-1 from being mis-folded and degraded, leading to more PDL-1 on the cell surface. The authors have done some site-mapping in a previous publication—this could be extended with proteomics to determine the % occupancy at each site before and after induction of EMT. With four sites, it is not at all clear what the requirements for “escape” from degradation are in terms of site occupancy. There are probably significant differences among different breast cancer cell lines and possibly differences between cancer stem cells and cancer non-stem cells. Con A binds to high-mannose, hybrid, and bi-antennary glycans and not to branched glycans, which would be predicted to be increased after EMT due to induction of specific glycosyltransferases. LC MS/MS analysis should demonstrate what is actually resulting from the EMT inducers.*

Response: We thank the reviewer for the comments. Our earlier studies [Nat Commun. 2016 Aug 30; 7:12632] demonstrated that PD-L1 is N-glycosylated at four sites (N35, N192, N200 and N219) and glycosylation at N192, N200 and N219 is critical for PD-L1 protein stabilization by preventing PD-L1 from ubiquitin/proteasome-mediated degradation (Figure 2 of the paper). Following the reviewer's suggestion, we performed mass spectrometry analysis with results showing that the N-glycan occupancy (%) of PD-L1 N-glycosylation sites was upregulated upon EMT, supporting our notion that EMT upregulates PD-L1 through PD-L1 glycosylation. The new results have been incorporated into **Supplementary Figure 3** of revised manuscript, and the experimental details are described below and added to the **Method** section of revised manuscript (**pages 34**).

To quantify the N-glycan occupancy (%) of each PD-L1 N-glycosylation site before and after EMT induction, total PD-L1 proteins, including glycosylated and non-glycosylated forms, were purified from epithelial and mesenchymal cells individually. To this end, Flag-PD-L1 was expressed in epithelial (untreated MCF-10A) or mesenchymal (TGF- β -treated MCF-10A) cells in the presence of proteasome inhibitor MG132 (to prevent non-glycosylated PD-L1 from degradation). Then, both glycosylated and non-glycosylated Flag-PD-L1 proteins were purified by Flag-beads and subjected to LC-MS/MS analysis.

N-glycan occupancy (%) of each N-glycosylation site was estimated based on the relative peak intensities of non-N-glycosylated and de-N-glycosylated peptides detected after removal of N-glycans by PNGase F. Enzymatic release of N-glycans would convert the originally occupied Asn (N) into Asp (D), creating a one mass unit difference from the corresponding peptide carrying a non-occupied site, which could be resolved by extracted ion chromatograms at 10 ppm accuracy based on the calculated accurate mass. To account for spontaneous deamidation as opposed to true enzymatic de-N-glycosylation, a control experiment was performed in which PNGase F was omitted. The intensities of two non-glycosylated tryptic peptides from PDL1, DQLSLGNAALQITDVK (m/z 843.4570) and AEVIWTSSDHQVLSGK (m/z 586.3003), were used to normalize the protein amount among samples. Assuming the recovery and MS response of a tryptic peptide carrying the NxT site relative to the same peptide carrying the de-N-glycosylated or deamidated DxT site is approximately the same, the % site occupancy can be calculated based on the formula $(INF-IND)/(IN+INF) \times 100$, in which INF = normalized intensity of peptide with DxT, after PNGase F treatment; IND = normalized intensity of peptide with DxT from the sample without PNGase F treatment (spontaneous deamidation); IN = normalized intensity of peptide with NxT from the sample treated with PNGase F. A 100% occupancy would imply IN = 0, with no spontaneous deamidation (IND = 0).

The extracted ion chromatograms for the PNGase F treated tryptic peptides corresponding to N35, N192 and N200 and their N-glycan site occupancy (%) were summarized below. Tryptic peptide carrying site N219 was not detected either as glycosylated or non-glycosylated peptide in this experiment, and thus its site occupancy (%) could not be estimated.

Supplementary Fig 3

a

NG site	Tryptic Peptide Sequence	Theoretical m/z of NxT	Theoretical m/z of DxT	N-glycan site occupancy (%)	
				Epithelial cells	Mesenchymal cells
35	K.DLYVVEYGS ^{N35} MTIECK.F	968.9369	969.4289	10.6	93.3
192	K.LFN ^{N192} VTSTLR.I	525.8007	526.2927	28.8	99.5
200	R.IN ^{N200} TTTNEIFYCTFR.R	890.4221	890.9141	38.1	99.7
219	R.RLDPEEN ^{N219} HTAELVIPELPLAHPNER.T	708.3646	708.6106	n/a	n/a

b

Quantification of N-glycan site occupancy of each PD-L1 N-glycosylation site upon EMT. (a) Table summarizing the N-glycan occupancy (%) of N35, N192 and N200 of PD-L1 purified from epithelial and mesenchymal cells. **(b)** The extracted ion chromatograms for the PNGase F treated tryptic peptides corresponding to N35, N192 and N200 of PD-L1 purified from epithelial and mesenchymal cells.

Point #3: In Line 188, it is not clear what, “enriches PD-L1 expression of CSCs” means.

Response: We apologize for the unclear description of the data. “enriches PDL1-expression of CSCs” means “induces more PD-L1 in CSCs than in non-CSCs”. In this paragraph, we would like to demonstrate that EMT induces more PD-L1 in CSCs than in non-CSCs through inducing higher STT3 in CSCs. To describe our conclusion more clearly, we modified the final sentence as follows:

In the original manuscript, the paragraph is: (the sentence mentioned in the comment is highlighted in yellow)

EMT-induced higher levels of STT3 in CSCs than in non-CSCs contributes to enriched PD-L1 expression of CSCs

The above-mentioned results prompted us to further ask whether STT3 isoforms may contribute to EMT-mediated enriched PD-L1 expression in CSCs than in non-CSCs. To this end, we compared EMT-mediated STT3 induction between CSC and non-CSC populations. The results showed that, in breast epithelial cells, while EMT driven by TGF- β or RasV12 upregulated STT3 isoforms in both populations, significantly higher levels of STT3 were observed in CSCs than in non-CSCs at both mRNA and protein levels (Fig. 3a,b). Higher STT3 expression was also detected in CSCs than in non-CSCs of breast cancer cells with intrinsic mesenchymal-like phenotype (Fig. 3c). Moreover, knockdown of both STT3 isoforms suppressed PD-L1 induction in both CSC and non-CSC populations and diminished EMT-mediated enriched PD-L1 expression in CSCs (Fig. 3d), leading to sensitization of CSCs to PBMC-mediated cancer cell killing (Fig. 3e). These results suggested that EMT induces higher levels of STT3 in CSCs than in non-CSCs and **enriches PD-L1 expression of CSCs.**

In the revised manuscript, the final sentence was modified to: (The modified portion is in yellow highlight)

EMT-induced higher levels of STT3 in CSCs than in non-CSCs contributes to enriched PD-L1 expression of CSCs

The above-mentioned results prompted us to further ask whether STT3 isoforms may contribute to EMT-mediated enriched PD-L1 expression in CSCs than in non-CSCs. To this end, we compared EMT-mediated STT3 induction between CSC and non-CSC populations. The results showed that, in breast epithelial cell MCF-10A, while EMT driven by TGF- β or RasV12 upregulated STT3 isoforms in both populations, significantly higher levels of STT3 were observed in CSCs than in non-CSCs at both mRNA and protein levels (Fig. 3a,b). Higher STT3 expression was also detected in CSCs than in non-CSCs of breast cancer cells with intrinsic mesenchymal-like phenotype (Fig. 3c). Moreover, knockdown of both STT3 isoforms suppressed PD-L1 induction in both CSC and non-CSC populations and diminished EMT-mediated enriched PD-L1 expression in CSCs (Fig. 3d), leading to sensitization of CSCs to PBMC-mediated cancer cell killing in vitro (Fig. 3e). These results suggested that EMT induces higher levels of STT3 in CSCs than in non-CSCs, **leading to enriched PD-L1 expression of CSCs.**

Point #4: *The effects of etoposide are interesting, but these effects should be the subject of a separate manuscript, after extensive documentation of the effects of the increases of endogenous STT3 on PDL-1 as well as control, cell surface glycoproteins.*

Response: We thank the reviewer for the suggestion. Because the other two reviewers also commented on these data, we will discuss with the editor regarding this suggestion and tentatively keep the revised manuscript as the original format.

Reviewer #3 (Remarks to the Author):

This is an interesting study showing that EMT triggers N-glycosylation of PD-L1, leading to enriched PD-L1 in CSCs via the b-catenin/STT3 axis. Previous studies by the same group indicate that the glycosylation of PD-L1 could stabilize PD-L1 expression on tumor cells and increase resistance to immune attack. These findings may help understand the mechanisms of CSCs in immune evasion. The paper is well-written and the conclusions are supported by the results in general.

Specific comments:

1. As we learn from numerous clinical studies, humanized anti-PD-1 or anti-PD-L1 antibodies can well saturate these molecules in vivo and provide occupancy of these targets at least a few weeks. The results presented in this paper may indicate a different way to down-regulate the expression of PD-L1 on cancer cells, including CSCs. While etoposide study presented here is interesting, this drug did not achieve the same efficacy as well as durability in cancer therapy as anti-PD-1/PD-L1 antibodies in clinic. Therefore, etoposide may only partially or less effectively modulate the expression of PD-L1 as its mechanism of action. In this context, the authors should carefully state the significance of their findings and discuss these possibilities.

2. The assays for cytotoxic T cells with the stimulation of human PBMCs activation in vitro by xenogeneic antigens and anti-CD3/IL-2 is highly artificial. While this assay indicates the change of cytotoxic activity of T cells, its significance is limited because this activity is much higher and well beyond normal T cell activity in real life. Therefore, I suggest that the authors to state the fact and do not use this data to implicate antitumor activity in vivo.

Authors' Response to Reviewer's Comments:

We are grateful to the reviewer for the comments and suggestions to improve the scientific merit of the manuscript. Below please find our point-by-point response to these comments.

Specific comment #1: *As we learn from numerous clinical studies, humanized anti-PD-1 or anti-PD-L1 antibodies can well saturate these molecules in vivo and provide occupancy of these targets at least a few weeks. The results presented in this paper may indicate a different way to down-regulate the expression of PD-L1 on cancer cells, including CSCs. While etoposide study presented here is interesting, this drug did not achieve the same efficacy as well as durability in cancer therapy as anti-PD-1/PD-L1 antibodies in clinic. Therefore, etoposide may only partially or less effectively modulate the expression of PD-L1 as its mechanism of action. In this context, the authors should carefully state the significance of their findings and discuss these possibilities.*

Response: We fully agree with the reviewer that in the clinic etoposide did not achieve the same therapeutic efficacy and durability as anti-PD-1/PD-L1 antibodies. We also agree that etoposide may modulate PD-L1 activity less effectively than anti-PD-L1 antibody as our results indicated that etoposide did not completely inhibit PD-L1 expression. Together, etoposide monotherapy is not comparable to anti-PD-1/PD-L1 antibodies in PD-1/PD-L1 immune checkpoint modulation and in cancer therapeutic efficacy and durability.

In revised manuscript, we toned down the regulatory efficacy of etoposide on PD-L1 expression and incorporated the reviewer's comments into the **Discussion** section of revised manuscript (**page 20, lines 10–14**) as follows:

“Although our results demonstrated that etoposide suppresses PD-L1, etoposide monotherapy does not achieve the same efficacy as well as durability in cancer therapy as anti-PD-1/PD-L1 antibodies in the clinic. This is likely attributed to the partial rather than complete downregulation of PD-L1 by etoposide, suggesting that etoposide monotherapy is not comparable to anti-PD-1/PD-L1 antibodies in PD-1/PD-L1 immune checkpoint modulation.”

Specific comment #2: *The assays for cytotoxic T cells with the stimulation of human PBMCs activation in vitro by xenogeneic antigens and anti-CD3/IL-2 is highly artificial. While this assay indicates the change of cytotoxic activity of T cells, its significance is limited because this activity is much higher and well beyond normal T cell activity in real life. Therefore, I suggest that the authors to state the fact and do not use this data to implicate antitumor activity in vivo.*

Response: We agree with the reviewer's comment that in vitro PBMC-mediated cancer cell killing assay cannot fully reflect the activity of T cells in vivo. Following the reviewer's suggestion, we modified our manuscript to clearly state that our PBMC-mediated cancer cell killing assays were performed in vitro and only described the results as in vitro findings.

In the original manuscript **from page 5, lines 4–10**, we described:

“We then compared the sensitivity of CSC and non-CSC populations to peripheral blood mononuclear cell (PBMC)-mediated cancer cell killing in the presence or absence of PD-L1. As expected, CSCs were more resistant to PBMC-mediated killing as shown by reduced level of cleaved caspase 3. However, following PD-L1 knockout, both CSC and non-CSC populations showed similar levels of cleaved caspase 3 (Supplementary Fig. 1d), indicating that enriched PD-L1-mediated CSC immune evasion can be recaptured in our breast cancer model system.”

In the revised manuscript, we modified the description according the reviewer's suggestion as shown below in yellow highlight:

“We then compared the sensitivity of CSC and non-CSC populations to peripheral blood mononuclear cell (PBMC)-mediated cancer cell killing **in vitro** in the presence or absence of PD-L1. As expected, CSCs were more resistant to PBMC-mediated killing **in vitro** as shown by reduced level of cleaved caspase 3. However, following PD-L1 knockout, both CSC and non-CSC populations showed similar levels of cleaved caspase 3 (Supplementary Fig. 1d), **suggesting that the enhanced PD-L1 expression in CSCs contributes to CSC resistance to PBMC-mediated killing in vitro in our breast cancer model system.**”

We also clearly described our PBMC killing assays as in vitro experiments in the following sentences:

Page 10, lines 2–5: Moreover, knockdown of both STT3 isoforms suppressed PD-L1 induction in both CSC and non-CSC populations and diminished EMT-mediated enriched PD-L1 expression in CSCs (Fig. 3d), leading to sensitization of CSCs to PBMC-mediated cancer cell killing **in vitro** (Fig. 3e).

Page 13, lines 16–18: Along with MET induction, etoposide attenuated glycosylated PD-L1 expression and PD-1 binding ability of sphere cells, and sensitized sphere cells to PBMC-mediated cancer cell killing **in vitro** (Fig. 6e and Supplementary Fig. 5b).

Page 16, lines 16–19: In addition, in line with the effects of etoposide treatment (Fig. 6d), TOP2B silencing induced MET and downregulated STT3 and PD-L1 in spheres (Fig. 8d, left), resulting in reduced PD-1 binding (Fig. 8d, upper right) and sensitization of sphere cells to PBMC-mediated cancer cell killing **in vitro** (Fig. 8d, bottom right).

REVIEWERS' COMMENTS:

Reviewer #1 (Remarks to the Author):

The authors have carefully and thoroughly addressed all the comments raised previously. The conclusions are solid and convincing. Most importantly, the findings in this study are significant and will be a great interest to the broad readership of cancer research community. Thus, the current version is in perfect shape for publication.

Reviewer #2 (Remarks to the Author):

The manuscript has been significantly improved with the removal of the tunicamycin data and the addition of site-mapping of N-linked glycan occupancy upon EMT. Perhaps after the extensive revisions made to address reviewer's comments, the Title should be revised.

Reviewer 3 commented for the editors only and was satisfied by the revision.

Point-by-Point Response to Reviewer's Comments

REVIEWERS' COMMENTS:

Authors' Response to Reviewer's Comments:

Reviewer #1 (Remarks to the Author):

The authors have carefully and thoroughly addressed all the comments raised previously. The conclusions are solid and convincing. Most importantly, the findings in this study are significant and will be a great interest to the broad readership of cancer research community. Thus, the current vision is in perfect shape for publication.

Response: We thank the reviewer for his/her time in reviewing our manuscript and the invaluable comments and suggestions.

Reviewer #2 (Remarks to the Author):

The manuscript has been significantly improved with the removal of the tunicamycin data and the addition of site-mapping of N-linked glycan occupancy upon EMT. Perhaps after the extensive revisions made to address reviewer's comments, the Title should be revised.

Response: We appreciate the reviewer's suggestion and have revised the title to "STT3-dependent PD-L1 accumulation on cancer stem cells promotes immune evasion".

Reviewer 3 commented for the editors only and was satisfied by the revision.

Response: We thank the reviewer for his/her time in reviewing our manuscript and the invaluable comments suggestions.